# Detecting Temporal Misalignment Attacks in Multimodal Fusion for Autonomous Driving

**Md Hasan Shahriar**[†]    **Md Mohaimin Al Barat**[†]    **Harshavardhan Sundar**[‡]
**Naren Ramakrishnan**[†]    **Ning Zhang**[§]    **Y. Thomas Hou**[†]    **Wenjing Lou**[†]
[†]Virginia Tech, Blacksburg, VA, USA
`{hshahriar, barat, naren, thou, wjlou}@vt.edu`
[‡]Amazon.com, Inc., Seattle, WA, USA
`hsundar427@gmail.com`
[§]Washington University in St. Louis, St. Louis, MO, USA
`zhang.ning@wustl.edu`

## Abstract

Multimodal fusion (MMF) is crucial for autonomous driving perception, combining camera and LiDAR streams for reliable scene understanding. However, its reliance on precise temporal synchronization introduces a vulnerability: adversaries can exploit network-induced delays to subtly misalign sensor streams, degrading MMF performance. To address this, we propose AION, a lightweight, plug-in defense tailored for the autonomous driving scenario. AION integrates continuity-aware contrastive learning to learn smooth multimodal representations and a DTW-based detection mechanism to trace temporal alignment paths and generate misalignment scores. AION demonstrates strong and consistent robustness against a wide range of temporal misalignment attacks on KITTI and nuScenes, achieving high average AUROC for camera-only (0.9493) and LiDAR-only (0.9495) attacks, while sustaining robust performance under joint cross-modal attacks (0.9195 on most attacks) with low false-positive rates across fusion backbones. Code is available at: `https://github.com/shahriar0651/AION`.

## 1 Introduction

Autonomous vehicles rely on multimodal fusion (MMF) of complementary sensors such as cameras and LiDAR to achieve robust perception (Zhang et al., 2023; Feng et al., 2020; Chen et al., 2017). While cameras provide rich semantic texture and LiDAR delivers accurate geometric depth, their integration crucially depends on precise temporal synchronization. Misalignments in frames can cause fusion models to miss objects or generate spurious detections, leading to significant safety hazards in downstream planning and control (Kuhse et al., 2024).

Recent studies, including our prior work, have shown that temporal desynchronization is not only a benign calibration issue but also a potential attack vector, known as a temporal misalignment (TMA) attack (Shahriar et al., 2026). Network-induced delays or timestamp manipulation can be exploited by adversaries to *misalign* sensor streams in time, thereby degrading the performance of detection and tracking without altering sensor content (Finkenzeller et al., 2025). Specifically, we demonstrated that even a single-frame LiDAR delay can reduce average precision by more than 88% across multiple detection models (Shahriar et al., 2026).

Existing efforts to handle temporal inconsistency primarily focus on calibration and benign jitter compensation, such as filtering or offline timestamp alignment (Taylor & Nieto, 2016; Zhao et al., 2021). While effective for clock drift or noise, these methods assume cooperative settings and do not detect deliberate, adversarial misalignments. On the defense side, most work has targeted adversarial examples or sensor spoofing (Sato et al., 2025; Gao et al., 2021), rely on spatial, semantic, or cross-modal inconsistencies through consistency checks, autoencoders, or hardware safeguards, leaving the temporal dimension of fusion largely unaddressed. Man et al. (2023) enforces track–label consistency but ignores timestamp validity; Li et al. (2020) detects context violations yet fails on time-shifted data; and Xu et al. (2024) catches gross spoofing but overlooks subtle desynchroniza-

tion within tolerance windows. To date, all defense mechanisms assume benign timestamps, leaving them vulnerable to network-level latency manipulation.

To address this gap, we propose AION, a lightweight defense patch that augments existing perception models by explicitly monitoring cross-modal temporal consistency. AION learns shared multimodal representations of camera and LiDAR inputs and applies dynamic time warping (DTW) to trace their temporal alignment path (Berndt & Clifford, 1994). In AD, consecutive frames are temporally adjacent and semantically similar, but standard contrastive learning treats pairs strictly as positive or negative. This rigid approach fails to capture subtle temporal misalignments. To address this, we introduce continuity-aware contrastive learning (CACL), which encourages the model to learn smooth temporal transitions. Specifically, we *estimate the "negativity" of two negative sample pairs based on their temporal distance*—pairs closer in time are penalized less than distant pairs—allowing the model to adaptively respect temporal continuity, enabling fine-grained multimodal representation.

Moreover, DTW is effective in analyzing temporal alignment because it does not assume uniform timing—a practical constraint for AD. Hence, instead of comparing sequences strictly index-to-index, DTW allows non-linear warping along the time axis, making it robust to delays, drifts, or jitter in multimodal sensors—precisely the distortions exploited by TMA attacks. Deviations in this alignment yield anomaly scores that indicate potential desynchronization or TMA attacks. In the absence of reliable network timestamps, AION leverages such semantic coherence between modalities to detect deviations in the time series input across different modalities.

Our contributions are as follows:

- We propose AION, a plug-in detection framework that couples multimodal representation learning with DTW-based temporal alignment and consistency monitoring, providing an efficient, downstream task-agnostic defense against TMA attacks.

- We introduce continuity-aware contrastive learning, which leverages temporal proximity to assign graded negativity to sample pairs, enabling the model to learn smooth temporal transitions and detect fine-grained misalignments in multimodal sensor data. We also demonstrate a novel use of DTW to *estimate temporal misalignment*, enabling real-time detection of subtle temporal manipulations.

- We introduce seven temporal misalignment (TMA) attacks covering both benign faults and adversarial patterns, and evaluate AION across multiple datasets and fusion backbones. It achieves high detection and defense performance with AUROC 0.9493 for camera-only, 0.9495 for LiDAR-only, and 0.9195 on most cross-modal attacks, while maintaining low false-positive rates, demonstrating robustness and generalizability. The implementation code and trained models are made public to ensure reproducibility.

## 2 BACKGROUND AND THREAT MODEL

**Dynamic Time Warping (DTW).** DTW is a classical technique for measuring similarity between two temporal sequences that may be out of phase or evolve at different speeds. Given sequences $X = (x_1, \ldots, x_n)$ and $Y = (y_1, \ldots, y_m)$, DTW computes a cost matrix $D(i,j) = d(x_i, y_j)$, where $d(\cdot, \cdot)$ is a local distance (e.g., Cosine, Euclidean, etc.). An alignment path is defined as $\mathcal{P} = \{(i_1, j_1), \ldots, (i_L, j_L)\}$, subject to boundary conditions $(i_1, j_1) = (1, 1)$, $(i_L, j_L) = (n, m)$, monotonicity, and continuity. The quality of a path is measured by its cumulative alignment cost:

$$C(\mathcal{P}) = \sum_{(i,j) \in \mathcal{P}} D(i,j),$$

and the optimal path is obtained as $\mathcal{P}^\star = \arg\min_{\mathcal{P}} C(\mathcal{P})$, which specifies how elements of $X$ and $Y$ should be aligned in time, while the minimal cost provides a quantitative measure of alignment quality—rewarding well-aligned sequences and penalizing distortions. This makes DTW a natural candidate for checking temporal alignment across multimodal signals that contain redundant information from the same surroundings.

**Temporal Synchronizer in AD.** We consider a multimodal perception pipeline for autonomous driving (AD) that fuses heterogeneous sensor modalities, focusing on camera ($S_C$) and LiDAR ($S_L$). At each discrete time step $t$, sensor $S \in \{S_C, S_L\}$ produces a message $\left(x_S^{(i)}, t_S^{(i)}\right)$, where

$x_S^{(i)}$ is the observation (image or point cloud) and $t_S^{(i)}$ is the sensor-reported timestamp. In most AD systems, sensor data are exchanged via middleware based on the Data Distribution Service (DDS). ROS 2, a widely used AD middleware, typically synchronizes cross-modal messages with an approximate-time synchronizer [1] that matches timestamps within a tolerance $\Delta t$. Concretely, each sensor modality $S$ keeps a finite FIFO buffer $\mathcal{Q}_S = \{m_{S,1}, \dots, m_{S,N}\}$ of recent messages (ordered by timestamp). An approximate-time synchronizer pairs messages across modalities based on timestamp proximity. For a new camera message (or LiDAR message), $m_C^{(i)}$, the synchronizer selects the LiDAR message (or camera message) with the closest timestamp,

$$j^\star(i) = \arg\min_k \left| t_C^{(i)} - t_L^{(k)} \right|,$$

and forms a pair $(m_C^{(i)}, m_L^{(j^\star)})$ if their reported time difference is within tolerance $\tau$ and that paired data is then processed and fused by the perception model.

**Multimodal Fusion-based Perception** Each modality has its own encoder $E_S$ that extracts feature-level representations: $f_C^{(i)} = E_C(x_C^{(i)})$ and $f_L^{(j^\star)} = E_L(x_L^{(j^\star)})$. The features are fused using a multimodal operator $F(\cdot)$, where $h^{(i)} = F\big(f_C^{(i)}, f_L^{(j^\star)}\big)$, and passed to a task-specific prediction head $g(\cdot)$, yielding the final output $y^{(i)} = g(h^{(i)})$. Thus, in the benign case, temporally aligned sensor data is paired, encoded, fused, and used to generate reliable perception outputs.

## 2.1 Threat Model

This part discusses the threat model, outlining how an adversary can exploit timestamp manipulation to disrupt sensor synchronization and compromise the perception pipeline (as outlined above).

**Attacker Objective.** We assume an adversary who does not tamper with raw sensor observations $x_S$ or the model parameters. Instead, the attacker manipulates the reported timestamps to force misaligned sensor pairs into the fusion stage. Concretely, for each message the adversary injects a perturbation $\delta_t^{(i)}$ such that the system receives $\tilde{t}_S^{(i)} = t_S^{(i)} + \delta_S^{(i)}$. The synchronizer then selects pairs according to manipulated timestamps,

$$\tilde{j}^\star(i) = \arg\min_k \left| \tilde{t}_C^{(i)} - \tilde{t}_L^{(k)} \right|,$$

resulting in fused features $\tilde{h}^{(i)} = F\big(E_C(x_C^{(i)}), E_L(x_L^{(\tilde{j}^\star)})\big)$. Even though the reported misalignment $|\tilde{t}_C^{(i)} - \tilde{t}_L^{(\tilde{j}^\star)}|$ is within tolerance $\tau$, the true temporal difference $\Delta_{\text{true}}^{(i,j)} = t_C^{(i)} - t_L^{(j)}$ may be large, producing semantically inconsistent feature pairs. These corrupted representations $\tilde{h}^{(i)}$ propagate through the fusion module, ultimately degrading predictions $\tilde{y}^{(i)}$ without requiring the attacker to alter raw sensor data or model parameters.

**Attacker capability.** We focus on the threat model where there is a compromised instance of in-vehicle ECU or the ROS2 middleware situated upstream of the fusion node. From this position, the attacker can read and write messages on the middleware bus and therefore inject messages $m_S^{(i)} = (x_S^{(i)}, \tilde{t}_S^{(i)})$, while leaving the payload $x_S^{(i)}$ untouched. This capability is practically plausible because many ROS2 deployments are not configured with authentication-by-default (Deng et al., 2022), and ECUs frequently run third-party or legacy software that enlarges the attack surface (Checkoway et al., 2011; Foster et al., 2015; Miller & Valasek, 2015; Yeasmin & Haque, 2021; Ghosal et al., 2022); a single compromised node, therefore, suffices to propagate forged timestamps to the fusion process. From an attacker's perspective, the objective is to corrupt the timestamps in a way that forces the approximate-time synchronizer to emit pairs for which true temporal separation $|\Delta_{\text{true}}^{(i,j)}| = \left| t_C^{(i)} - t_L^{(j)} \right|$ is large enough to break semantic correspondence and degrade downstream perception.

---

[1] `TimeSynchronizer` and `ApproximateTimeSynchronizer` are commonly used message filtering utilities in ROS2 that align multiple sensor message streams based on their timestamps. While `TimeSynchronizer` performs strict timestamp matching, `ApproximateTimeSynchronizer` allows messages with slight temporal differences—within a specified tolerance window—to be synchronized.

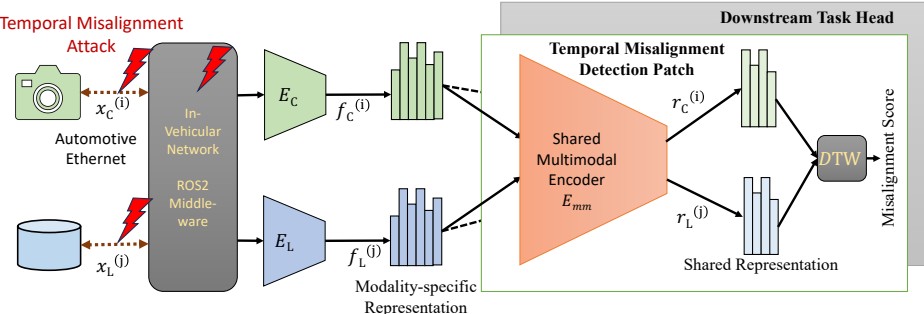

Figure 1: Overview of the proposed defense AION against any TMA attack.

**Defense Objectives.** A practical defense against temporal misalignment attacks must satisfy three key properties: i) it should accurately detect when sensor streams are out of sync, ii) generalize across different architectures and sensor modalities, and iii) introduce minimal overhead so that real-time perception pipelines remain unaffected. Meeting these requirements is essential for ensuring that AD systems remain both robust and deployable in practice.

## 3 TEMPORAL MISALIGNMENT DEFENSE: AION

To defend against such temporal misalignment attacks, we propose a countermeasure technique named AION, that can detect if any of the sensor data streams are misaligned. We design AION as an independent detection patch that can work on top of any MMF-based application, either in parallel or sequentially, agnostic of the downstream task.

### 3.1 AION OVERVIEW

As shown in Fig. 1, AION primarily consists of a single shared multimodal representation encoder (MRE) that maps any unimodal feature representation, regardless of its source or modality, to a shared representation space. Furthermore, AION has two phases of implementation: i) development and ii) deployment phase.

*Development Phase.* In the development phase, AION trains an MRE model using contrastive learning with positive and negative pairs based on their related temporal alignment. MRE learns how to represent temporally aligned (positive) feature pairs from different modalities to similar representations and temporally misaligned (negative) pairs to different representations. Once the MRE is trained, AION utilizes that trained MRE in the deployment phase to detect TMA attacks.

*Deployment Phase.* During the deployment phase, AION utilizes the trained MRE to create shared representations of historical inputs from each modality and keeps a stack of these representations for a small window. At the same time, AION also calculates and keeps track of a 2D similarity matrix with pairwise inter-modality similarity scores between different representation pairs. The diagonal elements in the similarity matrix indicate pairs that are temporally aligned and others that are temporally misaligned to different extents as they deviate from the diagonal. On each such similarity matrix, AION runs a dynamic time warping (DTW) algorithm to find the optimal path of temporal alignment and the reward of such alignment, which is the summation of all their similarity scores. Under a benign scenario, the optimal path with the highest reward would be the diagonal one, and the reward would be higher. However, under a temporal misalignment attack, the optimal path would deviate from the diagonal and follow the attacker's misaligned pattern. In that case, the optimal reward would be lower, which essentially indicates the existence of an adversary. We elaborate on the details of each component of AION in the following subsections.

### 3.2 TECHNICAL DETAILS OF AION

To learn a unified representation for multimodal inputs, we use a shared MRE, $E_{mm}$ that projects modality-specific features $f_C$ and $f_L$ from different modalities into a common latent space, such that $r_C^{(i)} = E_{mm}(f_C^{(i)})$ and $r_L^{(j)} = E_{mm}(f_L^{(j)})$. The objective is to ensure that the shared representations

of semantically corresponding (e.g., temporally aligned) inputs are close in the latent space, meaning $r_C^{(t_i)} = r_L^{(t_i)}$ if $i = j$, and dissimilar otherwise. As the majority of MMF–based perception models for AD primarily focus on fusing camera and LiDAR data, we center our technical discussion of AION on these two modalities.

The development phase specifically involves the training of the MRE model and running the detection on benign data to set the threshold. To ensure effective learning, we utilize contrastive learning with three types of data pairs for the model training.

### 3.2.1 DIFFERENT REPRESENTATION PAIRS.

To ensure that MRE effectively learns representations while respecting the subtle semantic changes in temporally adjacent frames, we categorize representation pairs into three types based on their degree of temporal (mis)alignment.

**Definition 1 (Positive Pairs)** *A pair of features* $(r_C^{(i)}, r_L^{(j)})$ *is called a* positive pair, *denoted* $(r_C^{(i)}, r_L^{(j)}) \in \mathcal{T}_p$, *if they originate from the same temporal event, i.e., $i = j$.*

**Definition 2 (Near-Negative Pairs)** *A pair* $(r_C^{(i)}, r_L^{(j)})$ *is called a* near-negative pair, *denoted* $(r_C^{(i)}, r_L^{(j)}) \in \mathcal{T}_{nn}$, *if they come from different but temporally adjacent events, i.e., $i \neq j$ but $i \approx j$. Such pairs share partially overlapping semantic content due to their temporal proximity.*

**Definition 3 (Far-Negative Pairs)** *A pair* $(r_C^{(i)}, r_L^{(j)})$ *is called a* far-negative pair, *denoted* $(r_C^{(i)}, r_L^{(j)}) \in \mathcal{T}_{fn}$, *if they originate from temporally distant events with no semantic overlap, i.e., $|i - j| \gg 0$.*

### 3.2.2 CONTINUITY-AWARE CONTRASTIVE LEARNING-BASED TRAINING

The primary goal of the shared encoder $E_{mm}$ is to ensure that the representations of *positive pairs* are highly similar—i.e., have minimal distance—while representations of *negative pairs* remain well separated in the latent space. To achieve this, we adopt a contrastive learning objective, based on relaxed contrastive (ReCo) as proposed in (Lin et al., 2023), to train $E_{mm}$, where each training batch consists of a set of discrete sample indices $\mathcal{I}_{batch} = \{n_1, n_2, \ldots, n_b\}$, where the batch size is $b$ and each $n_k$ corresponds to a unique sample in the batch.

Thus, the representation sequences $\mathbf{r}_C = \{r_C^{(n_1)}, r_C^{(n_2)}, \ldots, r_C^{(n_b)}\}$ and $\mathbf{r}_L = \{r_L^{(n_1)}, r_L^{(n_2)}, \ldots, r_L^{(n_b)}\}$ from two different modalities are calculated on the sampled inputs from the training set. These indices are chosen in a manner that ensures the batch contains both *near-negative* and *far-negative* pairs. Based on the $\mathbf{r}_C$ and $\mathbf{r}_L$, we compute a similarity matrix $\mathbf{S} \in \mathbb{R}^{b \times b}$, where each entry $S_{ij}$ denotes the cosine similarity between the camera representation $r_C^{(i)}$ and the LiDAR representation $r_L^{(j)}$, defined as:

$$S_{ij} = \frac{r_C^{(i)} \cdot r_L^{(j)}}{\|r_C^{(i)}\| \, \|r_L^{(j)}\|} \tag{1}$$

For *positive pairs*, we define the positive loss as: $\mathcal{L}_{\text{pos}} = \sum_{i=1}^{b} (S_{ii} - 1)^2$, which loss encourages the cosine similarity between the shared representations of temporally aligned inputs to be as close as possible to 1. *Negative pairs* consist of temporally misaligned inputs, and ideally, their representations should exhibit minimal cosine similarity. To enforce this, we define the negative loss as: $\mathcal{L}_{\text{neg}} = \sum_{\substack{i,j=1 \\ i \neq j}}^{b} (\max(0, S_{ij}))^2 \cdot \lambda_{ij}$. This loss penalizes any similarity between the negative pairs at different scales, which is the key enabler of CACL. The penalty is modulated by the weight $\lambda_{ij}$, which reflects the expected degree of dissimilarity based on temporal distance.

To generalize this weighting scheme, we define $\lambda_{ij}$ as a smooth function of temporal distance: $\lambda_{ij} = \tanh\left(\frac{|i-j|}{\tau}\right)$, where, $\tau$ is a temperature-like scaling factor that controls sensitivity to temporal separation. This formulation (as shown in Fig 5 in Appendix C) offers a continuous and differentiable measure of misalignment, encouraging the model to learn nuanced distinctions across the temporal spectrum. The overall objective combines the positive and negative pair losses, $\mathcal{L}_{\text{total}} = \mathcal{L}_{\text{pos}} + \mathcal{L}_{\text{neg}}$. This loss ensures high cosine similarity for aligned (positive) pairs, while

pushing apart misaligned (negative) pairs. The extent of separation for negative pairs is controlled by the penalty weight $\lambda_{ij}$, allowing for flexibility based on temporal misalignment.

## 3.3 ATTACK DETECTION

The detection of TMA attacks through AION consists of two main tasks.

### 3.3.1 HISTORICAL REPRESENTATION QUEUE AND SIMILARITY MATRIX

During testing, AION maintains a historical representation queue for each modality over a window of $w$ recent sensor readings. Rather than applying uniform sampling, AION employs an exponential sampling strategy. This approach samples densely near the current time and more sparsely further in the past, allowing the model to capture fine-grained recent temporal changes while retaining a longer historical context for better generalization. The sampling indices are defined as $n_i = \psi^i$, where $\psi$ is the sampling base, forming the detection index set $\mathcal{I}_{detect} = \{n_1, n_2, \ldots, n_w\}$. Across this window size $w$, AION tracks the sequential representations $\mathbf{r}_C = \{r_C^{(n_1)}, r_C^{(n_2)}, \ldots, r_C^{(n_w)}\}$ and $\mathbf{r}_L = \{r_L^{(n_1)}, r_L^{(n_2)}, \ldots, r_L^{(n_w)}\}$. Consistent with the training phase, AION uses these representations to construct the similarity matrix $\mathbf{S} \in \mathbb{R}^{w \times w}$ as defined in equation 1. Upon the arrival of each new message, AION dynamically updates $\mathbf{r}_C$, $\mathbf{r}_L$, and $\mathbf{S}$, and proceeds to execute the DTW-based detection process described in the following section.

### 3.3.2 DYNAMIC TIME WARPING-BASED DETECTION

To quantify the extent of temporal misalignment within the $\mathbf{r}_C$ and $\mathbf{r}_L$, AION employs DTW to compute both the optimal temporal alignment path and the corresponding alignment reward. We implemented DTW to identify the optimal warping path $\mathcal{P}$ that maximizes the accumulated similarity, which we define as reward, over a similarity matrix. Algorithm 1 (in Appendix B), outlines this procedure, which takes $\mathbf{S}$ as input and returns the optimal path $\mathcal{P}$ and total reward $\phi$ associated with that path. In an ideal scenario, where all sensors remain temporally aligned, the optimal warping path follows the diagonal: $\mathcal{P}^* = \{(1,1), (2,2), \ldots, (w,w)\}$, as diagonal elements $S_{ii}$ have the highest similarity scores. Under the optimal alignment path, the optimal accumulated reward, $\phi^* = \sum_{i=1}^{w} S_{ii} \approx w$, since the embedding function $E_{mm}$ is trained to maximize similarity for aligned pairs. Thus, any deviation from that diagonal path $\mathcal{P}^*$ or the optimal reward $\phi^*$ can be considered anomalous.

*Justification on Detection.* The fundamental assumption behind this approach is that DTW maximizes cumulative alignment reward by optimally aligning sequences. Given a well-trained $E_{mm}$, the cost function $S_{ij}$ satisfies:
$$S_{ij} \approx 1 \quad \text{iff} \quad i = j$$
In a benign case, where data samples are perfectly aligned, $\phi_{ben}$ is maximized, and $a_{ben}$ is minimized, since all elements on the optimal path mostly satisfy $i = j$, therefore:
$$\phi_{ben} = \sum_{(i,j) \in \mathcal{P}_{ben}} S_{ij} \approx \sum_{i=1}^{W} S_{ii} \quad \text{thus,} \quad a_{ben} \approx 0$$
However, in the presence of malicious misalignment, the warping path necessarily includes terms where $i \neq j$, leading to $S_{ij} << 1$ for some $(i,j)$. Since DTW maximizes the total reward, the deviation from $\mathcal{P}^*$ implies a decrease in $\phi_{mal}$ and an increase in $a_{mal}$, such that:
$$\phi_{mal} = \sum_{(i,j) \in \mathcal{P}_{mal}} S_{ij} < \sum_{i=1}^{W} S_{ii} \quad \text{thus,} \quad a_{mal} >> 0$$
This establishes the fundamental assumption that as misalignment increases, so does anomaly score, reinforcing the validity of DTW in the anomaly detection process. Empirical validation in Section 5.1 further supports this claim.

## 4 EXPERIMENTAL SETTINGS

To evaluate the effectiveness of AION in detecting TMA attacks, we conduct a detection analysis under various attack scenarios. We synthetically generate different degrees of temporal misalignment by perturbing the input sequences in the test data as described in Table 1. For two different

Table 1: Seven Temporal Misalignment Attack Strategies

| Attack | Type | Delay $\delta_S$ | Description |
|---|---|---|---|
| Constant | Freeze | $\delta_j = j$ (within window) | Frame freezing, dropped frames |
| Random | Replacement | Random from $[i - m, i]$ | Random replacements, corrupted frames |
| Jitter | Stochastic | $\delta_t = \mu + \varepsilon_t, \varepsilon_t \sim U(-\Delta, +\Delta)$ | Probabilistic jitter, network jitter |
| Reversal | Reordering | $\delta_j = 2j$ (within window) | Order reversal, out-of-order packets |
| Burst | Intermittent | $\delta_j = j$ (within burst) | Intermittent freezes, bursty congestion |
| Drift | Cumulative | $\delta_j = \lfloor r \times j \rfloor$ | Gradual desync, clock skew |
| Scheduler | Algorithm | $\delta_j = f(q, d_{\max})$ (round-robin/priority) | CPU scheduler (round-robin/priority) |

models trained on two different datasets, we evaluate AION's ability to distinguish between normal and misaligned sequences under diverse TMA attacks.

## 4.1 DATASETS

We evaluate AION on two standard multimodal AD datasets:

**KITTI Tracking Dataset.** The KITTI benchmark (Geiger et al., 2012), collected in Karlsruhe, Germany, covers city, residential, and highway scenes. It provides a forward-facing RGB camera and a Velodyne LiDAR, with 3D bounding boxes and labels for cars, pedestrians, and cyclists.

**NuScenes Dataset.** The NuScenes benchmark (Caesar et al., 2020), recorded in Boston and Singapore, captures dense urban traffic. It includes six RGB cameras, a Velodyne LiDAR, and five radars. NuScenes consists of 1000 20-second sequences with 3D bounding boxes and tracking IDs for different classes, such as vehicles, pedestrians, bicycles, and barriers.

## 4.2 MODEL ARCHITECTURE

We implemented AION for both the KITTI and nuScenes datasets to evaluate its adaptability across different sensor setups and driving scenarios.

**AION on KITTI:** For the KITTI dataset, we adopt a straightforward approach by testing with two off-the-shelf, pre-trained image and LiDAR feature encoders. The MRE of AION is implemented using a simple convolutional neural network (CNN) architecture, featuring two distinct input branches and a shared output branch. For each KITTI sample, an RGB image of size $[3, 375, 1242]$ is encoded using ResNet-50 (He et al., 2016) to produce image features $f_C \in \mathbb{R}^{2048 \times 12 \times 39}$, while the LiDAR point cloud $[k, 3]$ is processed by PointPillars (Lang et al., 2019) to yield LiDAR features $f_L \in \mathbb{R}^{384 \times 248 \times 216}$. Our encoder $E_{mm}$ maps both $f_C$ and $f_L$ to a shared space by applying modality-specific convolutional branches, global average pooling, and a shared projection head, producing 256-dimensional representations $r_C$ and $r_L$.

**AION on nuScenes:** For the nuScenes dataset, we build AION on top of BEVFusion (Liu et al., 2023) to demonstrate AION's adaptability to complex MMF architectures. Each input includes six camera images and a LiDAR point cloud. We use BEVFusion's encoders to obtain BEV features $f_C, f_L \in \mathbb{R}^{64 \times 180 \times 180}$ for camera and LiDAR, respectively. These are passed to a hybrid encoder $E_{mm}$, which first applies shared CNN layers to produce $[256 \times 23 \times 23]$ embeddings. A lightweight transformer then processes spatial tokens with positional encodings and global self-attention, followed by mean pooling to produce 256-dimensional representations $r_C$ and $r_L$.

## 4.3 EVALUATION SETTINGS

**Attack Hyperparameters.** We generate misaligned samples for both datasets using the TMA attacks defined in Table 1, injecting perturbations into test sequences at fixed intervals. All attacks follow a periodic structure parameterized by attack interval $t$ and duration $n$, set to default values $t = 25$ and $n = 10$ for consistent evaluation. Attack-specific parameters emulate realistic timing faults: Random uses a history window $m = 10$; Jitter applies mean delay $\mu = 1$ and jitter range $\Delta = 3$; Burst uses burst size $s = 3$ and gap $g = 2$; Drift applies drift rate $r = 0.5$; and Scheduler uses time quantum $q = 3$, maximum delay $d_{\max} = 5$, and round-robin scheduling. For multimodal attacks, we introduce controlled cross-sensor variability to model heterogeneous pipeline behavior, including $\pm 20\%$ variation for Random parameters $(t, n, m)$, $\pm 30\%$ variation for Jitter $(\mu, \Delta)$ and

Drift $(r)$, and $\pm 2$-frame variation for Burst $(s, g)$ and Scheduler $(q, d_{\max})$, along with phase offsets up to $40\%$ of the attack interval to enforce realistic temporal desynchronization between Camera and LiDAR streams.

**Anomaly Detection Methodology.** To classify whether an input sequence is malicious, we analyze the cross-modal temporal consistency of multimodal pairs within a defined observation window $w = 3$ with sampling base $\psi = 2$. In this evaluation, we label a window as *malicious* if at least half of its multimodal pairs contain a misaligned sample. We analyze the anomaly scores using the ROC curve and calculate the area under the ROC curve (AUROC) as the key detection metric.

**Software Implementation** We implement and evaluate AION in Python 3.8 using PyTorch, leveraging open-source frameworks such as OpenPCDet (Team, 2020). Experiments were primarily conducted on a high-performance server running Ubuntu 20.04.6 LTS, equipped with an Intel Xeon Gold 5520 CPU (16 cores, 2.20 GHz), 128 GB RAM, and three NVIDIA RTX 6000 Ada GPUs. For scalability profiling, we additionally evaluate AION on a more modest Ubuntu machine with an Intel Core i9-9820X CPU (10 cores, 3.30 GHz) and a single RTX 2080 Ti GPU, reflecting the constraints of a realistic automotive deployment scenario.

# 5 Detection Results

We evaluate the performance of AION across both datasets and model architectures. We begin by illustrating the detection process on the nuScenes dataset, including visualizations of similarity and anomaly scores under different attack types. Finally, we present the ROC curves, along with the AUROC scores, for both datasets.

## 5.1 Visualization of Similarity Matrix

Figure 2 illustrates four different similarity matrices with the Camera and LiDAR representations, $r_C$ and $r_L$, from time steps 10 to 40 under various TMA attacks (launched from 20 to 30), including the benign case. The top left-most panel shows the similarity matrix between $r_C$ and $r_L$ under benign conditions—i.e., with no delay in either modality. As illustrated, the highest similarity scores lie along the diagonal path from (10, 10) to (40, 40), indicating perfect temporal alignment between both modalities. However, all the following panels depict cases where seven types of temporal misalignments are introduced by delaying the camera stream under TMA attacks: all between time steps 20 and 30. In these scenarios, the highest similarity scores often diverge from the diagonal beyond time step 20 and only steadily return to the diagonal again around time step 30. These deviations clearly signify temporal misalignments, which AION leverages to detect such TMA attacks. We provide further figures in Appendix D, showing the impact of compromising only LiDAR (Figure 6) and both modalities (Figure 7).

Some plots in Fig. 7 show unique scenarios where both modalities are delayed by the same amount (*i.e.,* constant delay) under TMA attack. In this case, the similarity scores remain high (and the same) across both diagonal and occasionally, off-diagonal elements, from time steps 20 to 30. Such patterns may emerge under both benign and malicious conditions. For instance, under benign conditions, the vehicle may be stationary without any moving objects in the scene, resulting in temporally consistent features over time. In contrast, an attacker could also replicate this same scene with a malicious delay to all modalities by the same constant offset, creating a similar similarity matrix. Hence, these unique, advanced attacks become a challenging task just by analyzing the cross-modal alignment similarities. Although AION, when limited to only the modalities used in MMF, cannot reliably detect such an advanced attack case, incorporating additional data sources—such as inertial measurements (IMU), controller area network (CAN) signals, or other external references—can provide complementary evidence and help detect such advanced attacks. However, as we only rely on the multimodal data in this work, we consider this extension as future work for AION.

## 5.2 Detection Performance of AION

We illustrate the detection performance of AION from two different perspectives.

**Visualization of Anomaly Scores.** Figure 3 illustrates the temporal evolution of anomaly scores, provided by AION, across different time steps under various TMA attacks on the camera stream.

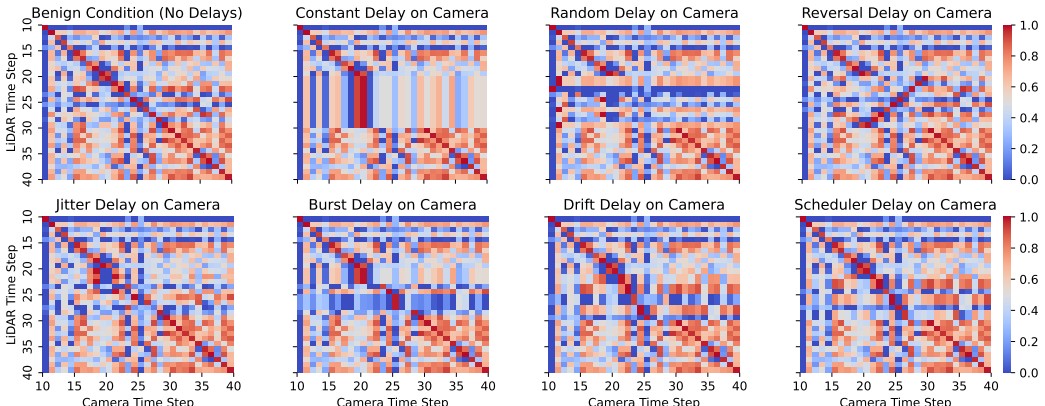

Figure 2: Similarity scores/matrix between Camera and LiDAR representation embeddings under different TMA attacks on Camera between time steps 20 to 30.

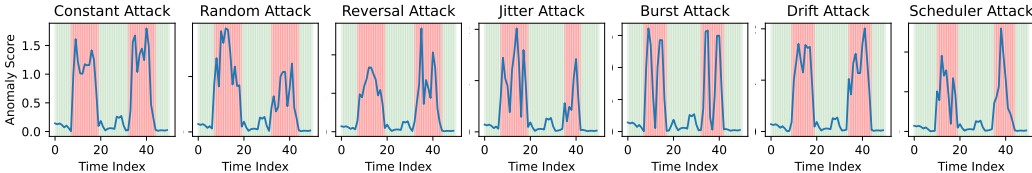

Figure 3: Anomaly scores generated by AION under various camera-only TMA attacks. The 'red' and 'green' shaded regions indicate periods with and without attacks, respectively. Distinctive score patterns across these regions highlight AION's effectiveness against diverse TMA attacks.

Each shaded region indicates whether the system is operating under benign (green) or malicious (red) conditions, based on the temporal alignment. As shown, AION consistently produces higher anomaly scores during periods with temporal misalignment than during benign intervals. This clear contrast in anomaly demonstrates the effectiveness of AION in detecting malicious temporal misalignment induced by a wide range of TMA attacks. Figure 8 in Appendix E shows the anomaly scores under LiDAR-based TMA attack.

**ROC Curve with AUROC Scores.** Figure 4 illustrates that across both datasets and in all *single-modality* settings, AION exhibits consistently strong detection performance. The ROC curves remain tightly concentrated near the top-left corner, and AUROC values range from 0.92–0.98. This robustness holds for all attack variants—including Drift, Jitter, Random, Burst, Scheduler, and even Reversal—demonstrating that AION reliably captures modality-specific temporal deviations without having dataset-specific dependency. High AUROC under Drift attacks further highlights AION's sensitivity to low-variance and slowly evolving perturbations, underscoring the strength of its DTW-based detection.

AION also maintains competitive performance under both-modality attacks for the majority of perturbations. Random, Drift, Jitter, Burst, and Scheduler attacks maintain AUROC scores close to their single-modality counterparts, indicating that AION effectively leverages cross-modal temporal correlations even when both sensors are perturbed. The close alignment of trends between KITTI and nuScenes suggests strong generalization across datasets with distinct motion statistics and sensor characteristics. Moreover, the consistently high true positive rates and low false positive rates underscore AION's reliability in realistic AD environments under TMA attacks.

A notable limitation arises under *perfectly synchronized, cross-modal* perturbations—such as Constant and Reversal attacks—when applied simultaneously to both camera and lidar. These attacks preserve the highest cross-modal similarity along the diagonal path, effectively suppressing the temporal discrepancies that AION relies upon for detection. This exposes an important avenue for future work by incorporating additional modalities (e.g., CAN, IMU, etc.) and developing invariant temporal anomaly features that remain robust under coordinated multi-sensor manipulation.

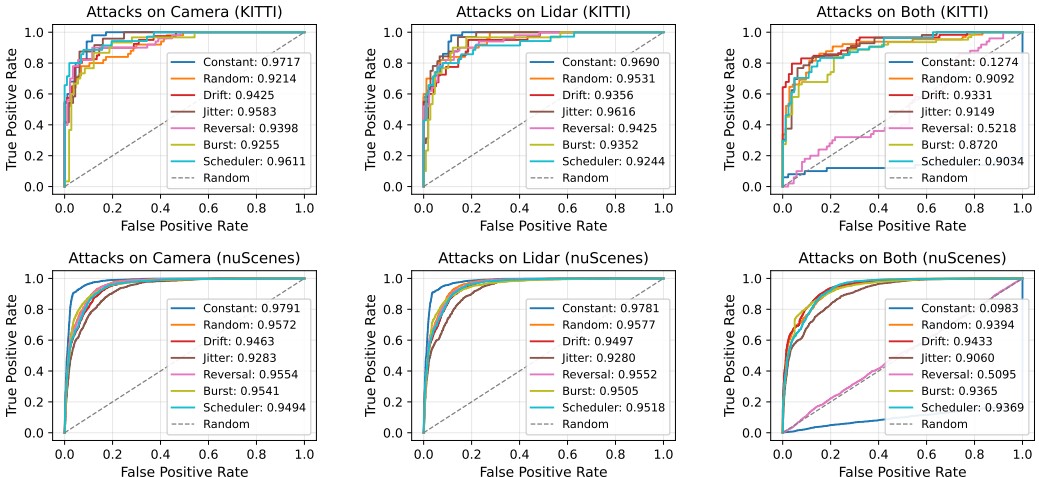

Figure 4: ROC curves with AUROC scores of AION under TMA attacks across KITTI and nuScenes, evaluated on camera, lidar, and both camera–lidar modalities.

**Scalability.** To enable efficient multimodal representation learning, AION introduces only a lightweight overhead. Below, we elaborate on the computation overhead of AION on the NuScenes dataset, as it is more computationally demanding with 6 cameras compared to the single front camera in KITTI. Compared to full perception model stacks, AION is highly compact, with only ∼1.97 million parameters (∼7.9 MB in FP32), whereas typical perception pipelines (such as BEVFusion) exceed 30 million parameters (∼127 MB in FP32) (Liu et al., 2023). On an RTX 2080Ti GPU, our profiling reveals that AION's computational overhead consists of two components: (1) MRE inference requires 1.74 ms per forward pass (574 inf/s) and consumes 42.5 MB of GPU memory, and (2) DTW-based detection adds 1.52 ms per inference (659 inf/s) and runs on CPU without GPU memory overhead. The total AION overhead is approximately 3.26 ms per inference with a combined throughput of ∼307 inf/s (see Table 2).

Given that typical MMF-based AD perception pipelines operate at 10–20 Hz (corresponding to 50–100 ms per frame), AION's overhead of 3.26 ms represents only 3.3–6.5% of the available frame budget, and can run in parallel to the downstream task. While DTW has $O(w^2)$ complexity, we empirically find that a short window

| Comp | Latency | Throughput | GPU Mem |
|------|---------|-----------|---------|
| MRE | 1.74 ms | 574 inf/s | 42.5 MB |
| DTW | 1.52 ms | 659 inf/s | – |
| **Total** | **3.26 ms** | **307 inf/s** | **42.5 MB** |

Table 2: Computational overhead of AION.

$(w = 3$ to $5)$ is sufficient to detect misalignment attacks in AD while keeping the runtime negligible and suitable for real-time deployment. Larger windows, on the other hand, add cost and may dilute temporal granularity, hurting effectiveness. This demonstrates that AION's robustness gains come at a very negligible computational cost, making it highly feasible for real-time deployment in production AD systems.

# 6 CONCLUSION

Temporal misalignment attacks are an emerging threat to AD perception, where adversaries manipulate timestamps—without altering sensor data—causing the temporal synchronizer to inadvertently induce cross-modal misalignment. To counter this challenge, we introduced AION, a lightweight defense that integrates multimodal representation learning with dynamic time warping to enforce temporal consistency before fusion. AION consistently exhibits strong robustness on diverse temporal misalignment attacks across KITTI and nuScenes, achieving high average AUROC scores for camera-only (0.9493) and LiDAR-only (0.9495) attacks, and maintaining resilient performance under both-modality attacks (0.9195 on most attacks). These results highlight the importance of synchronization-aware perception architectures and establish temporal consistency checking as a critical security property for safety-critical autonomous systems.

ACKNOWLEDGEMENTS

This work was supported in part by the Office of Naval Research under grants N00014-24-1-2730, N00014-24-1-2663; the National Science Foundation under Grants 2235232, 2312447, 2247560, 2154929, 2154930, 2238635, 2403758, 2509636, 2312794; the Army Research Office under grant W911NF-24-1-0155; and a fellowship from the Amazon–Virginia Tech Initiative for Efficient and Robust Machine Learning.

REPRODUCIBILITY STATEMENT

To facilitate reproducibility, we release our full implementation at `https://github.com/shahriar0651/AION`. The repository contains code for AION's MRE, CACL loss, DTW-based detector, and baseline methods, along with all configuration files and scripts needed to run the seven attack types and the hyperparameter sweep. We use publicly available KITTI and nuScenes datasets; preprocessing, feature extraction, and evaluation settings are detailed in Sections 5.1-5.2 and Appendices, and all reported AUROC and runtime results can be reproduced by running the provided evaluation scripts. We also provide the pretrained weights of MRE and the representation dataset for quick evaluation.

LLM USAGE DISCLOSURE.

LLMs were used for editorial purposes in this manuscript, and all outputs were inspected by the authors to ensure accuracy and originality.

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

## A    CONSIDERED ATTACKS

We evaluate our defense against seven distinct temporal misalignment attack strategies that model various real-world failure modes and adversarial scenarios in autonomous driving systems. Each attack operates on the temporal dimension of sensor data streams, introducing misalignment between camera and LiDAR modalities through different delay and manipulation patterns.

**Constant Attack**: Introduces complete frame freezing by replacing consecutive frames with a single frozen frame at periodic intervals. This attack simulates dropped frames or sensor failures, where frames within the attack window are replaced with the frame at the start of the window. The delay $\delta_j = j$ for position $j$ within the attack window, as each subsequent frame uses data from $j$ frames earlier, creating temporal plateaus that break cross-modal temporal alignment.

**Random Attack**: Applies random frame replacements by selecting frames uniformly at random from a history window $m$. This attack models corrupted frames or packet loss with random retransmission, introducing temporal discontinuities that disrupt the expected temporal ordering. Unlike delay-based attacks, this attack replaces frames with random historical frames rather than applying a deterministic delay distribution.

**Jitter Attack**: Applies stochastic timing delays following $\delta_t = \mu + \varepsilon_t$, where $\varepsilon_t \sim$ Uniform$(-\Delta, +\Delta)$ and $\delta_t$ represents the delay amount in frames. Each frame within the attack window is shifted backward by $\delta_t$ frames, modeling realistic network jitter and timing noise from scheduler fluctuations or multi-threading effects.

**Reversal Attack**: Reverses the temporal order of frames within periodic attack windows, where the delay $\delta_j = 2j$ for position $j$ within the attack window. This attack simulates out-of-order packet delivery scenarios, where frames arrive in reverse temporal sequence, violating the expected monotonic temporal progression.

**Burst Attack**: Introduces intermittent freezes with gaps between bursts, where frames within each burst are frozen by replacing them with earlier frames. This attack models bursty network congestion, where multiple short bursts of frozen frames are separated by gaps of normal operation. Within each burst, the delay increases as $\delta_j = j$ for position $j$ within the burst, creating a more subtle misalignment pattern than continuous freezing.

**Drift Attack**: Applies gradually accumulating delays following $\delta_i = \lfloor r \times i \rfloor$, where $r$ is the drift rate and $\delta_i$ represents the delay amount in frames. This attack simulates clock skew and buffer buildup scenarios, where delays accumulate linearly over time, either within periodic attack windows or continuously across all frames, creating a cumulative desynchronization effect.

**Scheduler Attack**: Mimics CPU scheduler behavior by applying delays following deterministic patterns $\delta_i = f(\text{quantum}, d_{\max})$, where $\delta_i$ represents the delay amount in frames and $f$ implements round-robin or priority-based scheduling algorithms. This attack models system-level delays from task preemption and priority-based processing, where delays cycle through values or increase based on priority aging mechanisms.

# B   DTW Algorithm for Aion in Misalignment Detection

Algorithm 1 summarizes how we apply DTW to measure temporal misalignment in AION and to quantify anomaly score on a window of multimodal representations.

---

**Algorithm 1:** Optimal Warping Path and Reward Computation

---

**Input**   : Cost matrix $\mathbf{S} \in \mathbb{R}^{w \times w}$
**Output:** Optimal path $\mathcal{P}^*$ and reward $\phi^*$
/* Initialization                                                              */
Initialize accumulated score matrix $R \in \mathbb{R}^{w \times w}$;
$R(1,1) \leftarrow S(1,1)$;
**for** $n \leftarrow 2$ **to** $w$ **do**
  | $R(n,1) \leftarrow S(n,1) + R(n-1,1)$;
**end**
**for** $m \leftarrow 2$ **to** $w$ **do**
  | $R(1,m) \leftarrow S(1,m) + R(1,m-1)$;
**end**
/* Dynamic programming recursion                                              */
**for** $n \leftarrow 2$ **to** $w$ **do**
  | **for** $m \leftarrow 2$ **to** $w$ **do**
  |   | $R(n,m) \leftarrow S(n,m) + \max\{R(n-1,m-1), R(n-1,m), R(n,m-1)\}$;
  | **end**
**end**
/* Backtracking                                                               */
$\mathcal{P}^* \leftarrow [(w,w)]$, $(n,m) \leftarrow (w,w)$;
**while** $(n,m) \neq (1,1)$ **do**
  | **if** $n = 1$ **then**
  |   | $m \leftarrow m - 1$;
  | **else if** $m = 1$ **then**
  |   | $n \leftarrow n - 1$;
  | **else**
  |   | $(n,m) \leftarrow \arg\max\{R(n-1,m-1), R(n-1,m), R(n,m-1)\}$;
  | **end**
  | Prepend $(n,m)$ to $\mathcal{P}^*$;
**end**
/* Final reward                                                               */
$\phi^* \leftarrow R(w,w)$;
**return** $\mathcal{P}^*$, $\phi^*$;

---

# C   Plot of $\lambda$ Function

In Fig. 5, we plot the $\lambda$ function to show how it assigns penalty weights based on the temporal misalignment $|i - j|$ between sequence elements, highlighting the effect of the sensitivity factor $\tau$ in controlling the transition from near-negative to far-negative pairs.

# D   Additional Figures on Cosine Similarity

We provide additional visualizations of the cosine similarity matrices under TMA attacks applied to both LiDAR-only (Figure 6) and both-modalities (Figure 7). These figures complement the main-text analysis by illustrating how TMA systematically perturbs the temporal similarity structure across different sensor configurations.

# E   Additional Visualization of Anomaly Scores

Figure 8 presents the anomaly score trajectories for LiDAR-based TMA attacks, complementing the camera-based results shown in Fig. 3. As with the camera stream, AION exhibits a clear separation between benign (green) and malicious (red) intervals, consistently producing elevated

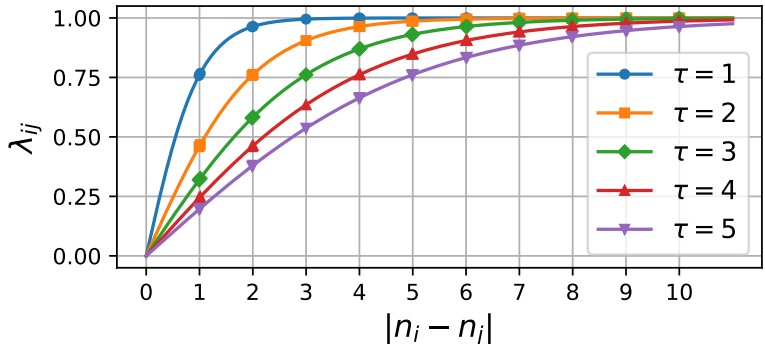

Figure 5: Visualization of the function $\lambda_{ij}$ for different misalignment level ($|i-j|$) and sensitivity factor ($\tau$). The x-axis represents the absolute difference $|i-j|$, indicating the transition from near-negative to far-negative pairs, and the y-axis shows the corresponding penalty weights $\lambda_{ij}$. Different lines indicate how the function saturates more quickly for smaller $\tau$, indicating the role of $\tau$ in setting the boundary between the near and far negative.

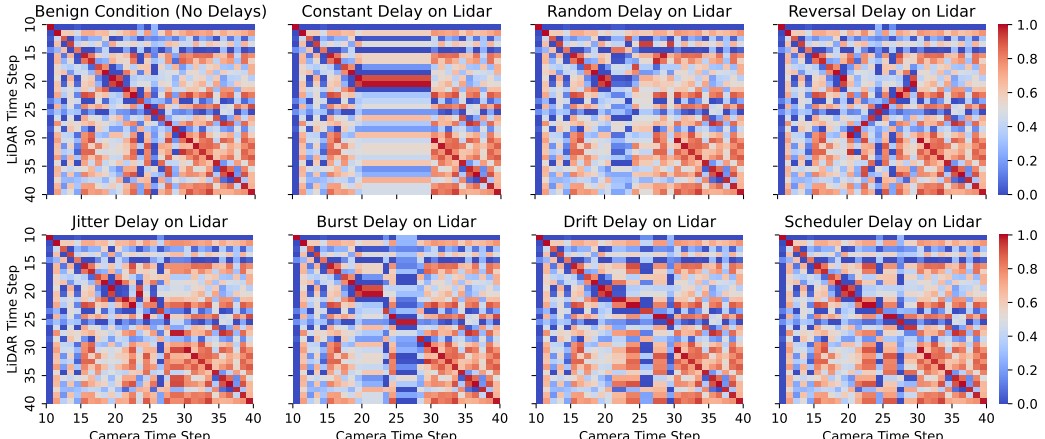

Figure 6: Similarity scores/matrix between Camera and LiDAR representation embeddings under different TMA attacks on LiDAR between time steps 20 to 30.

anomaly scores whenever temporal misalignment is introduced. This reinforces that AION detects misalignment-induced anomalies reliably across sensing modalities, including LiDAR.

## F    HARDWARE-IN-THE-LOOP TESTBED

To assess AION's compatibility with real-world deployment constraints, we develop a hardware-in-the-loop automotive Ethernet (AE) testbed that emulates a production in-vehicle sensing and fusion setup. As shown in Fig. 9, multiple Raspberry Pis, preloaded with the KITTI dataset, operate as camera and LiDAR sensor nodes, with an additional Pi and server CPU together serving as the fusion node. All these nodes communicate over AE using media converters and an AE switch. The full pipeline runs on ROS 2 over a data distribution service (DDS) for message distribution, enabling realistic sensor message timing and fusion workloads.

The camera and LiDAR nodes publish ROS 2 messages (images and point clouds) at a fixed rate (default $\approx 10\,\mathrm{Hz}$). Each message carries (i) the sensor payload and (ii) a header timestamp set from the node's wall clock at publish time. All nodes run in a single PTP domain and are synchronized with each other. The fusion node subscribes to both topics and runs ROS 2's `ApproximateTimeSynchronizer`: it maintains bounded per-topic queues and emits a camera–LiDAR tuple when the header timestamps of the two messages fall within a configurable

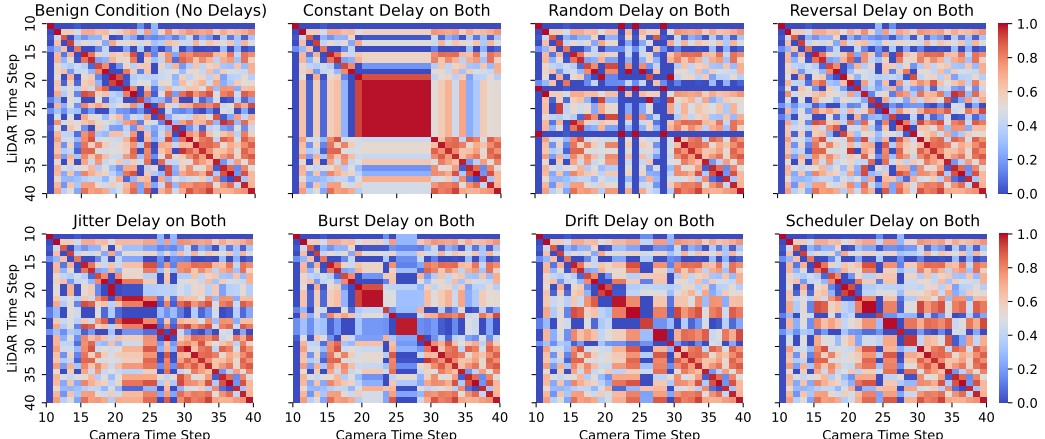

Figure 7: Similarity scores/matrix between Camera and LiDAR representation embeddings under different TMA attacks on Both Camera and LiDAR between time steps 20 to 30.

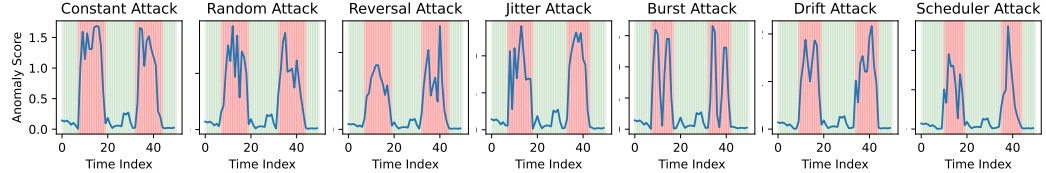

Figure 8: Anomaly scores generated by AION under various LiDAR-only TMA attacks. The 'red' and 'green' shaded regions indicate periods with and without attacks, respectively. Distinctive score patterns across these regions highlight AION's effectiveness against diverse TMA attacks.

tolerance (the slop $\tau = 0.10$ s). The fusion node then runs MMF-based perception on each emitted pair.

AION is integrated as a parallel callback to the perception stack that is also triggered whenever a new aligned sensor pair is selected for fusion. While the new data pair passes through the perception module, AION also updates its representation queue, computes a cross-modal similarity matrix, and performs DTW to evaluate the anomaly score. If the anomaly score falls within an acceptable threshold, the perceived result is released for further downstream tasks, such as planning and control. With this, the AION ensures that MMF-based autonomous driving is secure against TMA attacks.

We implement the temporal misalignment attack proposed in Shahriar et al. (2026), in which the adversary injects clock offsets by exploiting vulnerabilities in the PTP profile. Figure 11 illustrates how AION detects this attack. Under benign operation, the anomaly score remains consistently low, as shown in the figure. Once the attacker initiates the temporal offset injection, the anomaly score increases proportionally to the induced misalignment. After the attack ceases and synchronization stabilizes, the anomaly score gradually returns to its nominal low level. Theoretically, the anomaly score should remain close to 0, which can be obtained in the testbed with a proper calibration and configuration.

## G    SENSITIVITY ANALYSIS OF WINDOW SIZE

Figure 12 presents a sensitivity analysis evaluating the impact of the window size parameter $w \in \{3, 5, 7, 10\}$ on detection performance (AUROC) across various attack types for the KITTI and NuScenes datasets. The results demonstrate that moderate window sizes, specifically $w = 3$ and $w = 5$, consistently yield the highest detection accuracy across both datasets. As the window size increases to $w \geq 7$, we observe diminishing returns characterized by a general decline in AUROC, a trend that is particularly pronounced across all attacks in the NuScenes dataset and for the Scheduler

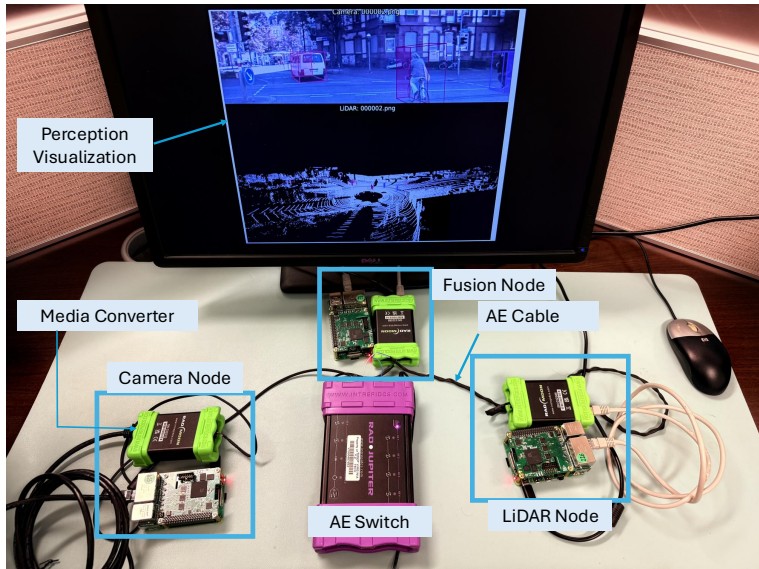

Figure 9: Hardware-in-the-loop (HIL) automotive Ethernet testbed

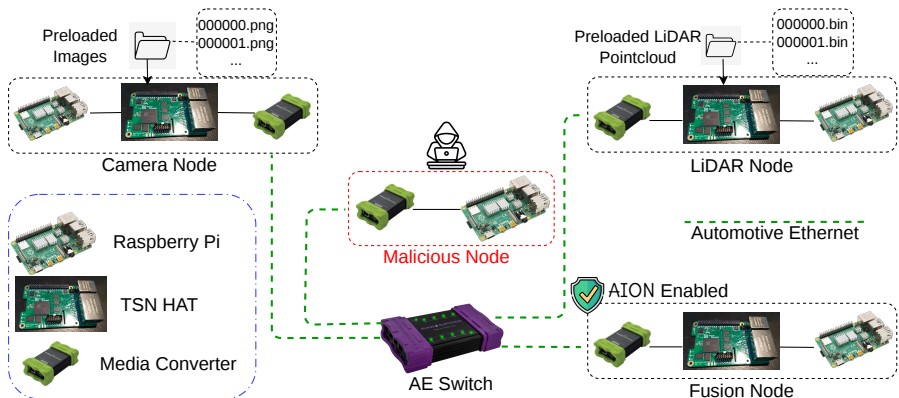

Figure 10: Schematic Diagram of HIL Testbed with AION enabled Fusion Node

attack in the KITTI dataset. Nevertheless, the extended evaluation highlights that the model's overall performance remains highly robust to the choice of $w$, maintaining strong detection capabilities provided the temporal window is not excessively long.

## H  THRESHOLD CALIBRATION AND TPR EVALUATION

To facilitate real-world deployment, we adopt a robust, single percentile-based global threshold calibrated exclusively on benign data. While Dynamic Time Warping (DTW) calculates the optimal alignment path reward, raw maximum similarity scores inherently fluctuate below 1 in practice due to scene- and context-specific variations. To mitigate this variance and ensure cross-context generalizability, we scale each similarity matrix to a $[0, 1]$ range. This normalization isolates the temporal alignment path from scene-specific limitations.

Under benign conditions, even if natural discrepancies exist between modalities due to external factors, the diagonal elements of the similarity matrix still reach their maximum values ($S_{i,i} \approx 1$), while off-diagonal elements remain significantly lower. Conversely, under a Temporal Misalignment Attack (TMA), specific off-diagonal elements artificially spike, forcing the DTW algorithm to deviate from the diagonal and penalizing the overall alignment reward. By evaluating benign segments, we

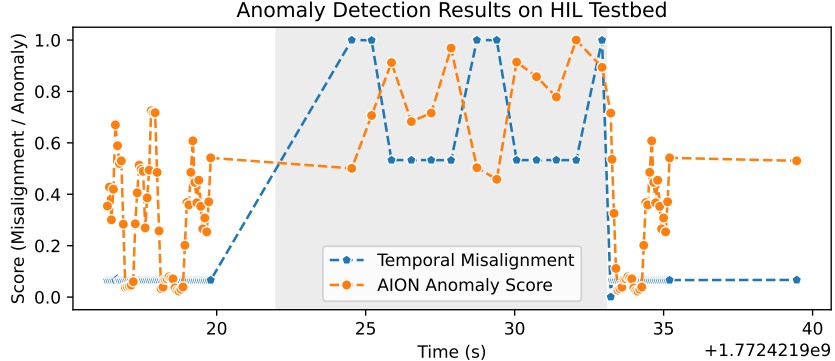

Figure 11: Anomaly score detection by AION in benign scenario and attack (shaded area) scenario

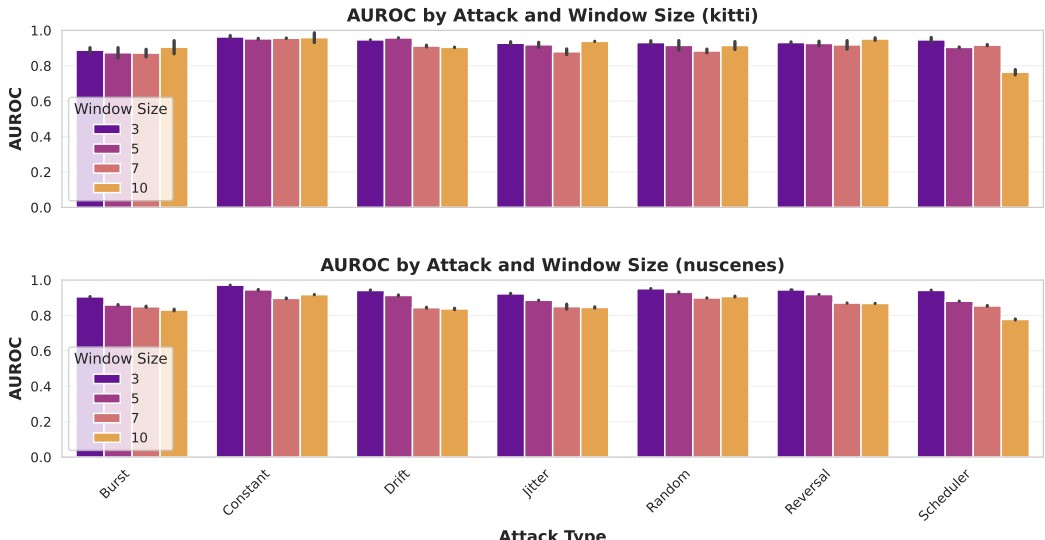

Figure 12: Sensitivity analysis of the impact of window size $w$ on detection performance (AUROC) across various attack types for the KITTI and NuScenes datasets.

establish a global threshold at a stringent high percentile (e.g., the $99^{\text{th}}$ percentile), effectively targeting a worst-case False Positive Rate (FPR) of $< 0.01$. This approach enables consistent, reliable deployment across diverse sequences and fleets without requiring continuous recalibration.

| Attack Type | KITTI | | | NuScenes | | |
| --- | --- | --- | --- | --- | --- | --- |
| | **Both** | **Camera** | **LiDAR** | **Both** | **Camera** | **LiDAR** |
| Burst | 0.2742 | 0.0333 | 0.2333 | 0.3307 | 0.3444 | 0.3333 |
| Constant | 0.0600 | 0.5400 | 0.5800 | 0.0033 | 0.4908 | 0.4942 |
| Drift | 0.6441 | 0.5500 | 0.5500 | 0.4221 | 0.3708 | 0.3656 |
| Jitter | 0.3036 | 0.4167 | 0.3125 | 0.2942 | 0.2486 | 0.2913 |
| Random | 0.3385 | 0.4000 | 0.6000 | 0.3688 | 0.3704 | 0.3488 |
| Reversal | 1.0000 | 0.4000 | 0.5200 | 0.0092 | 0.3538 | 0.3458 |
| Scheduler | 0.2963 | 0.7143 | 0.6000 | 0.3580 | 0.3643 | 0.3732 |

Table 3: True Positive Rate (TPR) at a False Positive Rate (FPR) of $< 0.01$ across various attack types and sensor modalities.

Table 3 presents the preliminary window-level True Positive Rate (TPR) at this target FPR. These metrics reflect window-level detection performance across the targeted modalities. Because some windows within a single extended attack event may exhibit only subtle misalignments, the per-window TPR is naturally constrained; however, the generalized thresholding strategy maintains high precision and strict false positive control.

## I BASELINE COMPARISON

To compare the performance of AION, we implement two temporal misalignment detection baselines: one based on timestamps and another on semantic content from the fused message pairs.

**Timestamp-sanity (interval variance).** This baseline operates purely on the sequence of timestamps and detects irregular timing patterns. Given a stream of timestamps $\{t_i\}$, we compute inter-arrival times $\Delta t_i = t_i - t_{i-1}$ and measure their normalized variance $\mathrm{var}(\Delta t)/(\mathbb{E}[\Delta t]^2 + \varepsilon)$. Under normal operation, messages arrive with nearly constant spacing, so the variance of $\Delta t$ is small. Manipulating timestamps or introducing abnormal jitter makes the gaps more irregular, increasing this variance. We use the resulting value as a scalar anomaly score and select a threshold based on benign data.

**Sliding-window correlation baseline.** This baseline quantifies the similarity between camera and LiDAR features at each time and interprets low similarity as evidence of misalignment. Let $r_C^{n_i}, r_L^{n_i} \in \mathbb{R}^d$ denote camera and LiDAR feature vectors at time step $n_i$, for a window of $w$. For each time step, we compute the Pearson correlation $\rho_{n_i} = \mathrm{corr}(r_C^{n_i}, r_L^{n_i})$ between the two feature vectors and then average these over time, $\bar{\rho} = \frac{1}{w} \sum_{i=1}^{w} \rho_i$. Well-synchronized streams yield feature pairs that are highly correlated (large $\bar{\rho}$), whereas temporal misalignment decorrelates the streams and decreases $\bar{\rho}$. We finally map this to an anomaly score via a simple linear transform, $(1 - \bar{\rho})/2$, and flag windows whose score exceeds a threshold calibrated on benign sequences.

### I.1 DETECTION PERFORMANCE

In Fig. 13, we compare AION to the two baseline detectors, *Timestamp* and *Correlation*, in terms of AUROC across all seven attack types on KITTI and nuScenes. On both datasets, AION consistently attains the highest or near-highest AUROC, typically in the 0.85–0.97 range, while the timestamp-sanity baseline lags behind (often below 0.8 on KITTI and 0.7 on nuScenes) and the correlation baseline sits in between. The gap is especially pronounced for the NuScenes dataset, where AION retains high AUROC, whereas the purely timestamp- or correlation-based methods degrade substantially. Overall, these results indicate that AION provides a significantly more reliable detector of temporal misalignment than either simple timestamp modeling or sliding-window feature correlation, and that its advantage is stable across datasets and attack families.

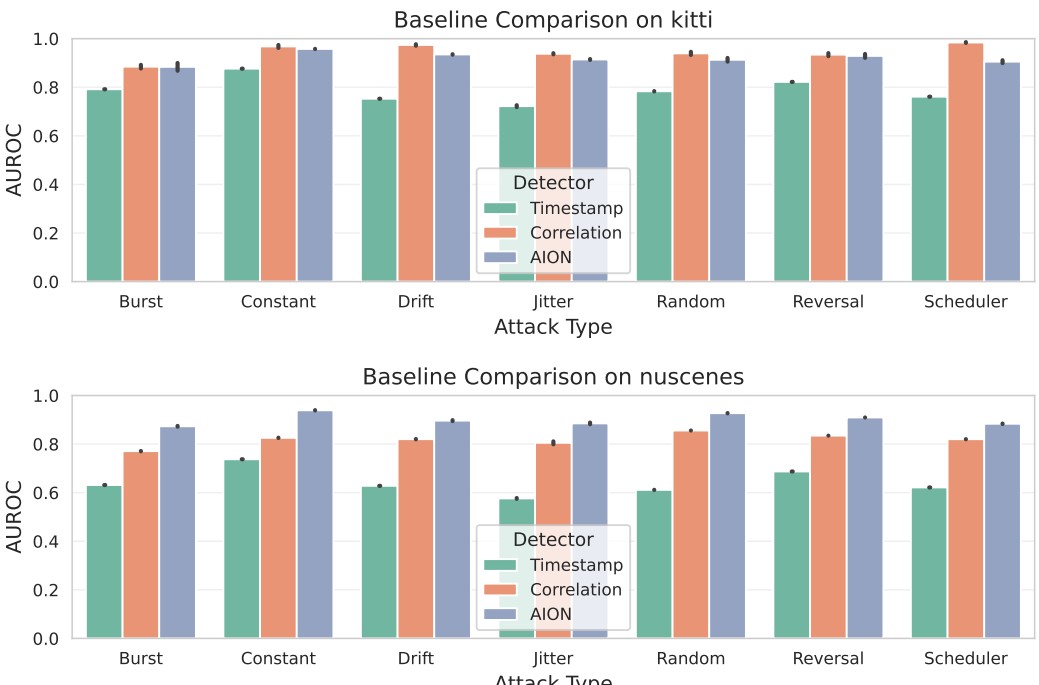

Figure 13: Baseline comparison of AUROC for the Timestamp and Correlation-based detectors across seven TMA attacks on KITTI (top) and nuScenes (bottom). AION consistently achieves higher AUROC than both baselines for most attack types, especially on complex datasets such as NuScenes, demonstrating its stronger robustness to diverse timing perturbations.

