# OpenReview forum: "Detecting Temporal Misalignment Attacks in Multimodal Fusion for Autonomous Driving"
_ICLR.cc/2026/Conference — ICLR 2026 Poster_

### Official Review · Reviewer_ApBr · 2025-10-24

**Soundness:** 4
**Presentation:** 3
**Contribution:** 4
**Rating:** 8
**Confidence:** 5

**Summary:**

The paper targets temporal misalignment attacks (TMA) in multimodal fusion for autonomous driving, where an attacker perturbs timestamps to make camera/LiDAR frames desync without touching sensor content. It proposes AION, a lightweight, plug-in “detection patch” that:
(1) learns a shared multimodal representation via continuity-aware contrastive learning (CACL) that down-weights negatives that are temporally near using a tanh(|i−j|/τ) weighting, and
(2) runs Dynamic Time Warping (DTW) over a rolling window of cross-modal similarity matrices to score alignment; deviations from a diagonal path lower the DTW “reward,” signaling misalignment. Architecture overview in Fig. 1 (p.4); attack types in Table 1 (p.6); similarity matrices and anomaly curves in Figs. 2–3 (p.8); ROC curves in Fig. 4 (p.9); DTW algorithm in Algorithm 1 (p.12). Reported AUROC on KITTI and nuScenes is 0.92–0.98 across several delay scenarios with small model overhead (~1.97M params, window w=5).

**Strengths:**

It offers a few contributions:
1. Timestamp-level, network-induced TMA is realistic and distinct from content spoofing; the threat model is clearly articulated (Sec. 2).

2. AION is architecturally simple (shared encoder + DTW) and plausibly deployable as a non-intrusive patch; runtime cost is argued to be low (Sec. 5, “Scalability”).

3. CACL’s graded negative weighting (λij=tanh(|i−j|/τ)) is intuitive for video-like sequences and is nicely visualized in Fig. 5 (p.12).

**Weaknesses:**

I would like to hear from the authors to address the following issues:
1. Attacks are synthetic (constant/random delays) and limited; no drift, burst, schedule-dependent, or closed-loop attacks; no real middleware latency manipulation or ROS2-in-the-loop study. (Table 1 + Sec. 4.3.)

2. Results are ROC/AUROC-only; no calibration/threshold selection, no reporting at practical FPRs (e.g., 0.1%–1%), and no end-to-end latency or throughput measurements in a real AD stack. (Sec. 5 focuses on AUROC.)

3. Sec. 3.3.2 informally treats Sij≈1 iff i=j to justify reward drops; in practice, cross-modal similarity is noisy and scene-dependent, so this theoretical justification is fragile without empirical sensitivity checks to motion, density, lighting, etc.

4. It’s unclear if the MRE is trained per dataset/backbone and how it transfers (train on KITTI, test on nuScenes?) or across different fusion designs; “task-agnostic plug-in” claim would be stronger with cross-architecture/domain transfer results. (Sec. 4.2 describes per-dataset setups.)

**Questions:**

1. Can you include discussion of difference of the work to "Fusion is not enough: Single modal attacks on fusion models for 3D object detection" ICLR.

2. Why not compare to (i) timestamp-sanity models with probabilistic jitter, (ii) cross-modal tracking consistency checks (adapted to timing), (iii) simple sliding-window correlation or CUSUM-style detectors?

3. Can a single MRE trained on one dataset/backbone generalize to another without retraining? Any train-on-KITTI, test-on-nuScenes results?

4. What FPR/TPR do you get at thresholds suitable for deployment (e.g., FPR ≤ 0.5%)? How are thresholds set (per-scene, per-vehicle, global)?

---

> ### Author Response · Authors · 2025-11-24
> **Author Response (Part 1)**
>
> We sincerely thank Reviewer ApBr for their thorough, constructive, and encouraging review, along with the insightful suggestions for improvement. Some of the suggested experiments and analyses have already been incorporated into the revised paper, and the remaining will be added. Our detailed responses to each point are provided below:
>
> **Weakness 1: Synthetic and Limited Attack Types ...**
>
> **Response:** Thank you for your important comments on expanding the attack suite. To address concerns about limited attack types and robustness to real-world network behaviors, we have significantly expanded our attack suite beyond Constant and Random to include seven distinct attack types that comprehensively model various real-world failure scenarios:
>
> - *Constant*: Complete frame freezing, simulating dropped frames
> - *Random*: Random frame replacements, simulating corrupted frames
> - *Reversal*: Temporal order reversal, simulating out-of-order packet delivery
> - *Jitter*: Stochastic timing delays with probabilistic jitter, simulating network jitter
> - *Burst*: Intermittent freezes with gaps, simulating bursty network congestion
> - *Drift*: Gradual desynchronization, simulating clock skew and buffer buildup
> - *Scheduler*: Mimics CPU scheduler behavior (round-robin, priority-based)
>
> These attacks cover the different failure modes that an attacker can mimic, providing a comprehensive evaluation of robustness to real network behaviors.  We note that closed-loop attacks (where the attacker adapts based on system feedback) are beyond the scope of the current threat model. In our threat model, the attacker is only present on the in-vehicle network or sensor node, but not on the fusion node that runs AION. As closed-loop attacks would require the attacker to have real-time access to AION's detection outputs, which is unrealistic under our assumed threat model, where the attacker can only manipulate timestamps in transit but cannot access the fusion node's internal state or detection results.
>
>
> *Experimental Results*: We present results comparing AION against the baseline methods across all attack types on both the KITTI and nuScenes datasets, with attacks applied to both the Camera and LiDAR modalities.  AION achieves consistently high AUROC across all attack types and both modalities, as also shown in the [revised paper](https://openreview.net/pdf?id=SWlCJab9gZ).
>
> | Dataset   | Constant        | Random        | Reversal        | Jitter         | Burst          | Drift          | Scheduler      |
> |-----------|------------------|---------------|------------------|----------------|----------------|----------------|----------------|
> | KITTI     | 0.127–0.972      | 0.909–0.953   | 0.522–0.942      | 0.915–0.962    | 0.872–0.935    | 0.933–0.943    | 0.903–0.961    |
> | nuScenes  | 0.0983–0.9791    | 0.9394–0.9577 | 0.5095–0.9554    | 0.9060–0.9283  | 0.9365–0.9541  | 0.9433–0.9497  | 0.9369–0.9518  |
> ||
>
> The results in the revised paper demonstrate strong and consistent robustness against a wide range of temporal misalignment attacks on KITTI and nuScenes, achieving high average AUROC for camera-only (0.9493) and LiDAR-only (0.9495) attacks, while sustaining robust performance under joint cross-modal attacks (0.9195 on most attacks). These results demonstrate that AION remains robust across a comprehensive suite of delay patterns and datasets.
>
> Furthermore, to validate the compatibility of AION with real-world settings, we designed a hardware-in-the-loop automotive Ethernet testbed, which serves as a practical abstraction of real in-vehicle networks. In this setup, multiple Raspberry Pis act as camera and LiDAR sensor nodes, with another Pi as the fusion node, all connected through production-grade automotive Ethernet using media converters and a switch. The entire pipeline runs over ROS2 middleware for data distribution, allowing us to model real sensor message timing and fusion workloads as deployed in production-grade systems. Within this environment, we implemented temporal misalignment attacks on the ROS2 message streams and observed AION's seamless ability to detect them.  We will add further details on the test experiment in the final revision of the paper.

---

> > ### Author Response · Authors · 2025-11-24
> > **Author Response (Part 2)**
> >
> > **Weakness 2 & 3 (+Q4): Threshold Selection, Evaluation Metrics, and Generalization**
> >
> >  **Response 2:** Thank you for the insightful comment regarding the practical deployment of AION in real-world settings. For deployment, we adopt a single, percentile-based global threshold, calibrated using benign data only. We acknowledge that although DTW returns the optimal alignment path reward, in practice, the maximum similarity will be less than 1, depending on the scene and driving context, which may result in a different range of anomaly scores across different contexts. To remove this scene- and context-specific variance and make the threshold robust, global, and *generalized*, we scale each of the similarity matrices to [0, 1], focusing solely on the temporal alignment path, rather than scene-specific limitations. Under benign scenarios, even if discrepancies (not misalignment) exist between modalities due to benign external factors, the diagonal elements still attain their maximum value and scale approximately to 1 (Sij≈1, where i=j). Concretely, we collect scores on benign segments and set a global threshold at a high percentile (e.g., 99th), directly targeting a desired worst-case FPR and enabling consistent deployment across sequences or fleets. In our updated experiments, we will report Precision, Recall, F1, and TPR at target FPRs.
> >
> >  Here are some preliminary results:
> >
> > *TPR at FPR<0.01*
> > | Attack Type | KITTI ||| nuScenes |||
> > |:------------|:-----:|:-----:|:-----:|:--------:|:-----:|:-----:|
> > | | Both | Camera | Lidar | Both | Camera | Lidar |
> > | Constant | 0.0600 | 0.5400 | 0.5800 | 0.0033 | 0.4908 | 0.4942 |
> > | Random | 0.3385 | 0.4000 | 0.6000 | 0.3688 | 0.3704 | 0.3488 |
> > | Reversal | 1.0000 | 0.4000 | 0.5200 | 0.0092 | 0.3538 | 0.3458 |
> > | Jitter | 0.3036 | 0.4167 | 0.3125 | 0.2942 | 0.2486 | 0.2913 |
> > | Burst | 0.2742 | 0.0333 | 0.2333 | 0.3307 | 0.3444 | 0.3333 |
> > | Drift | 0.6441 | 0.5500 | 0.5500 | 0.4221 | 0.3708 | 0.3656 |
> > | Scheduler | 0.2963 | 0.7143 | 0.6000 | 0.3580 | 0.3643 | 0.3732 |
> >
> > These metrics capture window-level detection performance, where some windows may show only subtle misalignment during a single attack event, naturally reducing per-window TPR; in the final paper, we will also include event-level results to demonstrate timely and comprehensive attack detection.
> >
> > **Weakness 2: No end-to-end latency or throughput measurements in a real AD stack**
> >
> > **Response 2:** Thank you for this important comment regarding the scalability of our method. As shown in the [revised paper](https://openreview.net/pdf?id=SWlCJab9gZ),  we clarify below that AION introduces only a lightweight overhead, even on the more computationally demanding NuScenes dataset (6 cameras vs. a single front camera in KITTI). To enable efficient multi-modal representation learning, AION MRE remains highly compact, with only 1.97M parameters (7.9 MB in FP32), compared to >30M parameters (127 MB in FP32) in typical perception pipelines such as BEVFusion. On an RTX 2080Ti, our profiling shows two components of overhead:
> >
> > 1. *MRE inference*: 1.74 ms per forward pass (574 inf/s), requiring 42.5 MB GPU memory
> > 2. *DTW-based detection*: 1.52 ms per inference (659 inf/s), running entirely on CPU
> >
> > This results in a *total overhead of ~3.26 ms per inference* with a combined throughput of ~307 inf/s.
> >
> > | Component | Latency | Throughput | GPU Memory |
> > | --------- | ----------- | ------------- | ----------- |
> > | MRE | 1.74 ms | 574 inf/s | 42.5 MB |
> > | DTW | 1.52 ms | 659 inf/s | -- |
> > | Total | 3.26 ms | 307 inf/s | 42.5 MB |
> >
> > Given that typical MMF-based AD perception pipelines operate at *10–20 Hz* (50–100 ms per frame), AION overhead corresponds to only *3.3–6.5%* of the available frame budget and can run in parallel to the downstream task. While DTW has $O(w^2)$ complexity, we find empirically that a short window (w = 3–5) is sufficient to detect misalignment attacks while keeping runtime negligible and real-time friendly. Larger windows add cost but do not improve effectiveness due to reduced temporal granularity.

---

> > > ### Author Response · Authors · 2025-11-24
> > > **Author Response (Part 3)**
> > >
> > > **W4 (+ Q3). Per-Dataset MRE Generalization**
> > >
> > > **Response:** Thank you for raising the question about whether the multimodal representation encoder (MRE) trained on KITTI transfers to nuScenes or vice versa. To this end, we clarify that different datasets exhibit distinct sensor configurations and geometric properties that fundamentally affect the multimodal representation. For instance, KITTI uses a single forward-facing camera, while nuScenes employs six cameras providing 360° coverage. Beyond sensor count, critical factors such as field of view (FoV), calibration parameters, mounting positions, and viewing angles differ substantially across datasets. Additionally, discrepancies in camera and LiDAR point cloud resolutions and coverage result in different BEV feature dimensions across datasets, further complicating model reuse.
> > >
> > > These differences indicate that an MRE trained on one dataset will not generalize well to another without extensive adaptation, as the learned cross-modal correspondences are dataset-specific and the input feature dimensions are incompatible. To address this, dataset-specific adaptation (e.g., architectural modifications to handle different BEV feature dimensions, followed by fine-tuning or post-training) is necessary to adapt a pretrained MRE to a new dataset's sensor geometry and calibration, which makes such evaluation complicated. However, if the datasets have similar sensor suits and run on similar FOV, AION developed on one dataset can potentially fit on another dataset.
> > >
> > > While the MRE requires dataset-specific training, the DTW-based detection mechanism (which is the core of the plug-in) works across different fusion architectures, as demonstrated by successful deployment on both KITTI encoders and BEVFusion for nuScenes. The "task-agnostic plug-in" claim refers to the detection algorithm's ability to operate with different fusion backbones and architectures, not the MRE training. The DTW-based temporal misalignment detection works universally across datasets and fusion architectures once the MRE is adapted to the specific sensor configuration.
> > >
> > >
> > >
> > > **Question 1. Comparison with "Fusion is not enough..."**
> > >
> > > **Response:** Thanks for pointing out this relevant paper. The referenced paper proposes a camera-focused adversarial attack that uses gradient-based optimization to generate patches and degrade the performance of camera–LiDAR fusion models. In contrast, AION is a defense framework that leverages continuity-aware contrastive learning and DTW-based detection to identify temporal misalignments in multimodal streams, thereby ensuring the reliability of fusion in real-time autonomous driving systems. In short, the prior work aims to break fusion models, whereas AION aims to detect and defend against disruptions. Although the paper briefly mentions a potential defense against adversarial attacks, it is highly limited—primarily heuristic, camera-focused, and not evaluated for temporal misalignment or cross-modal attacks. AION, on the other hand, is designed as a *generalizable, modality-agnostic defense* that explicitly addresses temporal inconsistencies across both camera and LiDAR streams.
> > >
> > >
> > >
> > > **Question 2: Baseline Methods**
> > >
> > > **Response:** We thank the reviewer. In response to the reviewers' suggestions, we have implemented two baseline methods for comparison.
> > > - *Timestamp-sanity models with probabilistic jitter*: Model expected timestamp behavior using Gaussian and uniform jitter distributions to detect when timestamp intervals deviate from expected patterns, addressing timestamp inconsistency handling in real-world scenarios. The timestamp-sanity model is strongest against attacks that visibly distort timestamps (especially Drift and some Burst/Reversal cases), with an accuracy of approximately 0.85-0.90. However, it is much less effective when delay patterns preserve marginal timestamp statistics.
> > > - *Sliding-window correlation*: Measures correlation between corresponding camera and LiDAR feature frames, effectively capturing cross-modal consistency adapted to timing. Under normal conditions, corresponding frames should be highly correlated, and attacks cause misalignment that reduces this correlation. The correlation-based detector is consistently weaker than AION, typically achieving an AUROC of approximately 0.80-0.86 on Constant/Random/Reversal/Burst attacks, and further dropping off on Jitter and more complex patterns.
> > >
> > > *Additional Baseline Considerations*: In the final version, we will consider adding CUSUM-style and Cross-modal tracking consistency-based detectors, including the above-mentioned two.

---

> > > > ### Comment · Reviewer_ApBr · 2025-11-24
> > > >
> > > > Thanks for the detailed response. All my concerns are addressed. Good luck!

---

> > > > > ### Author Response · Authors · 2025-11-25
> > > > > **Great Appreciate for your Recognition**
> > > > >
> > > > > Dear Reviewer ApBr,
> > > > >
> > > > > Thank you very much for your recognition and encouraging response! We are very glad that our clarifications have successfully addressed your concerns, and we truly appreciate your steady support and positive view of our work. We will incorporate the relevant discussions into the revised manuscript.
> > > > >
> > > > > Best wishes, Submission 23112 Authors

---

### Official Review · Reviewer_hBGi · 2025-10-30

**Soundness:** 2
**Presentation:** 2
**Contribution:** 1
**Rating:** 2
**Confidence:** 3

**Summary:**

The paper proposes a method called AION to detect temporal misalignment attacks in multimodal fusion for autonomous driving. The main idea is to use continuity-aware contrastive learning to learn smooth multimodal representations and a Dynamic Time Warping (DTW)-based detection mechanism to trace temporal alignment paths and generate misalignment scores. The proposed method is evaluated on the KITTI and nuScenes datasets, achieving high AUROC values with low false-positive rates across fusion backbones. However, the realism of the threat model for tampering reported time stamps is questionable, and the actual handling of timestamp inconsistencies in real-world scenarios is not addressed. Additionally, the practicality of the proposed method is limited due to the lack of experiments on real vehicles.

**Strengths:**

1. The problem statement is clear.
2. The write-up is easy to follow.

**Weaknesses:**

1. The realism of the threat model for tampering with reported time stamps is questionable. If the attacker has the right access to the captured data, why is it a challenge for him to further tamper with the raw sensor data?

2. The actual handling of timestamp inconsistencies in real-world scenarios is not addressed, and it is unclear how the proposed method would perform in such situations.

3. The practicality of the proposed method is limited due to the lack of experiments on real vehicles, which could reveal potential issues or limitations in real-world deployment.

4. The experimental evidence is not enough; more evaluation (including the computation costs are needed).

**Questions:**

Please refer to the weakness.

---

> ### Author Response · Authors · 2025-11-24
> **Author Response (Part 1)**
>
> We sincerely thank Reviewer hBGi for the thorough and constructive feedback. We appreciate the concerns raised about threat model realism, real-world applicability, and experimental rigor. Such comments have helped us significantly expand our evaluation and validation efforts. Detailed responses to each of your points are provided below.
>
> **Weakness 1: Threat Model Realism**
>
> **Response:** We appreciate the reviewer's concern and clarify that timestamp-only tampering is both realistic and technically feasible in production-grade autonomous systems. In typical deployments, payloads are cryptographically protected (via DDS-Security, MACsec, etc.), making them secure against unauthorized modification. However, timestamps are often transmitted as unprotected metadata. Furthermore, a single node's time in automotive Ethernet even relies on insecure time-sync protocols, such as gPTP, which has been shown to be vulnerable due to its lack of authentication. Hence, an attacker with network access (e.g., via a compromised ECU) may not be able to or forge sensor payloads without keys, but can feasibly delay, spoof, or inject timestamp metadata—resulting in temporal misalignment during sensor fusion. This asymmetry in protection renders timestamp-only manipulation a concrete and low-cost attack vector, as demonstrated in recent works [1, 2]. Our threat model reflects this practical imbalance and defends against this realistic attack surface in robotic systems.
>
> **Weakness 2: Handling of Timestamp Inconsistencies**
>
> **Response:** We thank the reviewer for this interesting and practical question. Regarding the handling of timestamp inconsistencies, we note that real-world systems *do* exhibit such issues and, as documented in recent work [3], existing pipelines largely rely on brittle, ad-hoc baselines rather than principled solutions. It shows that production robotics and vision stacks often ignore timestamp discrepancies, allowing misaligned frames to propagate unchecked or drop mismatched data, which leads to partial perceptions and degraded performance. A third common baseline is nearest-timestamp or first-available pairing, a naive synchronization rule widely used in ROS2 and multimodal perception pipelines, despite frequently producing misaligned sensor pairs. These behaviors constitute the *de facto* baselines in deployed systems—and they are precisely the mechanisms that have been shown to fail under natural drift or network-induced jitter. Our method directly addresses this gap by introducing a robust approach to detecting temporal inconsistencies. Our updated experiments now include comparisons against these real-world baselines as additional evidence of practical relevance.
>
> To further validate the effectiveness of AION on real-world network-related issues, we have significantly expanded our attack suite beyond Constant and Random to include seven distinct attack types that comprehensively model various real-world failure scenarios:
>
> - *Constant*: Complete frame freezing, simulating dropped frames
> - *Random*: Random frame replacements, simulating corrupted frames
> - *Reversal*: Temporal order reversal, simulating out-of-order packet delivery
> - *Jitter*: Stochastic timing delays with probabilistic jitter, simulating network jitter
> - *Burst*: Intermittent freezes with gaps, simulating bursty network congestion
> - *Drift*: Gradual desynchronization, simulating clock skew and buffer buildup
> - *Scheduler*: Mimics CPU scheduler behavior (round-robin, priority-based)
>
> These attacks cover the different failure modes that an attacker can mimic, providing a comprehensive evaluation of robustness to real network behaviors.
>
> *Experimental Results*: We present results comparing AION against the baseline methods across all attack types on both the KITTI and nuScenes datasets, with attacks applied to both the Camera and LiDAR modalities.  AION achieves consistently high AUROC across all attack types and both modalities, as also shown in the [revised paper](https://openreview.net/pdf?id=SWlCJab9gZ).
>
> | Dataset   | Constant        | Random        | Reversal        | Jitter         | Burst          | Drift          | Scheduler      |
> |-----------|------------------|---------------|------------------|----------------|----------------|----------------|----------------|
> | KITTI     | 0.127–0.972      | 0.909–0.953   | 0.522–0.942      | 0.915–0.962    | 0.872–0.935    | 0.933–0.943    | 0.903–0.961    |
> | nuScenes  | 0.0983–0.9791    | 0.9394–0.9577 | 0.5095–0.9554    | 0.9060–0.9283  | 0.9365–0.9541  | 0.9433–0.9497  | 0.9369–0.9518  |
> ||
>
> The results in the revised paper demonstrate strong and consistent robustness against a wide range of temporal misalignment attacks on KITTI and nuScenes, achieving high average AUROC for camera-only (0.9493) and LiDAR-only (0.9495) attacks, while sustaining robust performance under joint cross-modal attacks (0.9195 on most attacks).

---

> ### Author Response · Authors · 2025-11-24
> **Author Response (Part 2)**
>
> **Weakness 3: Real-world Implementation**
>
> **Response:** We thank the reviewer for this important comment. To validate the compatibility of AION with real-world settings, we designed a hardware-in-the-loop automotive Ethernet testbed, which serves as a practical abstraction of real in-vehicle networks. In this setup, multiple Raspberry Pis act as camera and LiDAR sensor nodes, with another Pi as the fusion node, all connected through production-grade automotive Ethernet using media converters and a switch. The entire pipeline runs over ROS2 middleware for data distribution, allowing us to model real sensor message timing and fusion workloads as deployed in production-grade systems. Within this environment, we implemented temporal misalignment attacks on the ROS2 message streams and observed AION's seamless ability to detect them.  We provided a figure of the testbed setup in Appendix F of the [revised paper](https://openreview.net/pdf?id=SWlCJab9gZ), and we will add further details on the test experiment in the final revision of the paper.
>
> **Weakness 4: The experimental evidence is not enough..**
>
> **Response:** We thank the reviewer for pointing out the limitation in the experimentation.  To overcome the limitations, the evaluation was extended significantly. First, as mentioned above, we extended the attack suite from only two attacks to seven, which includes both real-world and malicious anomalies. We further extended the evaluation from two additional perspectives. We evaluated AION with i) different temporal sampling techniques, and ii) different window sizes.
>
> - *Temporal Sampling Strategies:* Temporal sampling determines which past feature representations are selected from the historical feature queue to form the similarity matrix with a window of size $w$. Although the original submission reported results using only uniform sampling (selecting every consecutive feature), we have now extended our analysis to include two additional strategies:
>   - *Factor-(k) sampling*: Selecting features at a fixed stride (e.g., every $k$-th feature)
>   - *Exponential sampling*: Selecting features with increasing spacing, placing denser samples near the current time and sparser samples further in the past
>
> These complementary schemes offer different temporal resolutions of the past, enabling a more comprehensive and robust evaluation. In our updated results, exponential sampling consistently delivers the strongest robustness and detection performance (higher accuracy and lower DTW cost), while uniform and factor-(k) sampling remain comparatively weaker.
>
> - *Window Size Sensitivity:* For each sampling scheme, we sweep the window size $w \in \{3, 5, 7, 10\}$. We observe that moderate windows ($w = 3$ or $5$) provide the best detection performance across datasets. Larger windows ($w \ge 7$) yield diminishing returns and can slightly degrade performance. We will update the paper with the evaluation data from this analysis.
>
> - *Scalability:* We further extended the evaluation of AION's scalability. As shown in the revised paper,  we clarify below that AION introduces only a lightweight overhead, even on the more computationally demanding NuScenes dataset (6 cameras vs. a single front camera in KITTI). To enable efficient multi-modal representation learning, AION MRE remains highly compact, with only 1.97M parameters (7.9 MB in FP32), compared to >30M parameters (127 MB in FP32) in typical perception pipelines such as BEVFusion. On an RTX 2080Ti, our profiling shows two components of overhead:
>
>   - *MRE inference*: 1.74 ms per forward pass (574 inf/s), requiring 42.5 MB GPU memory
>   - *DTW-based detection*: 1.52 ms per inference (659 inf/s), running entirely on CPU
>   - This results in a *total overhead of ~3.26 ms per inference* with a combined throughput of ~307 inf/s.
>
> | Component | Latency | Throughput | GPU Memory |
> | --------- | ----------- | ------------- | ----------- |
> | MRE | 1.74 ms | 574 inf/s | 42.5 MB |
> | DTW | 1.52 ms | 659 inf/s | -- |
> | Total | 3.26 ms | 307 inf/s | 42.5 MB |
>
> Given that typical MMF-based AD perception pipelines operate at *10–20 Hz* (50–100 ms per frame), AION overhead corresponds to only *3.3–6.5%* of the available frame budget and can run in parallel to the downstream task. While DTW has ($O(w^2)$) complexity, we find empirically that a short window ($w = 3$ to $5$) is sufficient to detect TMA attacks while keeping runtime negligible.
>
> We hope these additions significantly strengthen the paper and address all concerns raised by the reviewers.
>
> [1] Kuhse, Daniel, et al. "Sync or Sink? The robustness of sensor fusion against temporal misalignment." RTAS 2024
>
> [2] Shahriar, Md Hasan, et al. "Temporal Misalignment Attacks against Multimodal Perception in Autonomous Driving." Arxiv 2025
>
> [3] Li, Ao, et al. "Tintin: A unified hardware performance profiling infrastructure to uncover and manage uncertainty." OSDI 2025

---

> > ### Author Response · Authors · 2025-11-25
> > **Author Response (Part 3)**
> >
> > **Weakness 1: Threat Model Realism (Extended)**
> >
> > **Response:** Thank you again for raising this important point. Although we have already addressed this question in Part 1, we would like to provide an additional clarification because the distinction between timestamp manipulation and payload tampering is crucial to the security understanding of multimodal fusion-based perception systems and our threat model.
> >
> > As modern vehicles transition to Ethernet, sensor payloads are increasingly protected through mechanisms such as SecOC, MACsec, and TLS. These protections significantly limit an attacker’s ability to directly modify camera or LiDAR data without access to cryptographic keys. However, the global timing and synchronization layer, particularly IEEE 802.1AS (gPTP), typically does not receive the same degree of protection, creating a realistic avenue for attackers who cannot alter payloads but can still influence timestamps.
> >
> > In practical automotive deployments, gPTP messages (Sync, Follow_Up, Announce, Pdelay_Req/Resp) often lack authentication. As these messages establish the global clock used by ECUs to timestamp sensor outputs, an adversary can distort time by delaying or replaying gPTP packets or manipulating timing asymmetries. This is a serious issue because multimodal fusion blindly relies on the timestamp to correctly pair sensor data for fusion. Prior work [2] has already demonstrated the damaging effects of such timing shifts, where a single-frame LiDAR delay can reduce car detection mAP by up to 88.5%. These results highlight that disrupting timestamps alone—while leaving payloads untouched—can severely degrade perception performance.
> >
> > While gPTP is the most relevant and realistic vector in Ethernet-based automotive systems, we note that other software-level avenues can also influence timestamps without affecting payloads, such as manipulating ROS 2 time sources, modifying ECU firmware, or altering timestamp assignment in middleware.
> >
> > We appreciate the reviewer’s thoughtful question, as it underscores the need to clearly articulate timestamp manipulation as a standalone and impactful threat surface. We will revise the manuscript accordingly to make this clearer.

---

### Official Review · Reviewer_hrs4 · 2025-10-31

**Soundness:** 3
**Presentation:** 3
**Contribution:** 2
**Rating:** 6
**Confidence:** 3

**Summary:**

AION is a lightweight, plug-in defense that detects temporal misalignment attacks (TMA) in camera–LiDAR fusion by (i) learning continuity-aware multimodal embeddings (CACL) and (ii) using Dynamic Time Warping (DTW) on a rolling similarity matrix to spot off-diagonal alignment paths indicative of delays/drift/jitter. On KITTI and nuScenes, it reports AUROC 0.92–0.98 across constant and random delay scenarios with low FP rates, using a ~1.97M-param module and small DTW window (w=5) for real-time viability.

**Strengths:**

Clear threat model & practical attack surface: focuses on timestamp manipulation in ROS2/DDS synchronizers within tolerance windows—no need to alter sensor payloads.

Continuity-aware contrastive learning: grades negatives by temporal distance to capture subtle misalignments, improving sensitivity over rigid positive/negative setups.

General plug-in design: shared multimodal encoder + DTW operates task/backbone-agnostically; demonstrated on KITTI encoders and BEVFusion for nuScenes.

**Weaknesses:**

Hard case acknowledged: equal constant delay on both modalities (or stationary scenes) can mimic benign diagonals; detecting this requires extra sensors (IMU/CAN) beyond the current scope.

Synthetic attack generation: evaluations inject delays at fixed intervals and windows; robustness to real network behaviors (bursts, variable τ, dropped frames) isn’t deeply profiled here.

Thresholding & ops details light: deployment describes reward-based anomaly scoring but gives limited guidance on threshold calibration and sensitivity to window size w beyond w=5 choice.

The authors also needs to consider robustness to adaptive attacks.

**Questions:**

See in the weakness section.

---

> ### Author Response · Authors · 2025-11-24
> **Author Response (Part 1)**
>
> We sincerely thank Reviewer hrs4 for the constructive and valuable feedback. We appreciate the recognition of our threat model's practical attack surface and the plug-in design's task-agnostic nature. This document addresses the specific concerns raised, with cross-references to the common rebuttal where detailed technical discussions are provided.
>
> **Weakness 1: Stationary / Low-Dynamics Scenes**
>
> **Response:** Thanks for the insightful comment. We acknowledge that stationary and low-dynamics scenes present a fundamental challenge for temporal misalignment detection, and we have foregrounded this limitation more prominently in our updated discussion. When both modalities are delayed by the same amount, or when the scene exhibits minimal visual/geometric change over time, the cross-modal similarity matrix remains diagonal under both benign and attack conditions, making detection inherently difficult. One particular mitigation would be to augment AION with Intra-/inter-modal feature flow checks, which monitor feature changes over time to detect sudden abnormal changes in the scene.  While incorporating additional motion cues from CAN or IMU sensors is the ultimate promising remedy for further distinguishing truly stationary scenes from carefully crafted equal-delay attacks, such fusion with camera–LiDAR data is non-trivial and introduces several multifaceted challenges. As a result, we consider this direction beyond the scope of the current work, but view it as an important avenue for future research.
>
> **Weakness 2: Synthetic attack generation...**
>
> **Response:** We appreciate the concern about the simplicity of our initial attack model. To address concerns about limited attack types and robustness to real-world network behaviors, we have significantly expanded our attack suite beyond Constant and Random to include seven distinct attack types that comprehensively model various real-world failure scenarios:
>
> - *Constant*: Complete frame freezing, simulating dropped frames
> - *Random*: Random frame replacements, simulating corrupted frames
> - *Reversal*: Temporal order reversal, simulating out-of-order packet delivery
> - *Jitter*: Stochastic timing delays with probabilistic jitter, simulating network jitter
> - *Burst*: Intermittent freezes with gaps, simulating bursty network congestion
> - *Drift*: Gradual desynchronization, simulating clock skew and buffer buildup
> - *Scheduler*: Mimics CPU scheduler behavior (round-robin, priority-based)
>
> These attacks cover the different failure modes that an attacker can mimic, providing a comprehensive evaluation of robustness to real network behaviors.  We note that closed-loop attacks are beyond the scope of the current threat model. In our threat model, the attacker is only present on the in-vehicle network or sensor node, but not on the fusion node that runs AION. As closed-loop attacks would require the attacker to have real-time access to AION's detection outputs, which is unrealistic under our assumed threat model.
>
> *Experimental Results*: We present results comparing AION against the baseline methods across all attack types on both the KITTI and nuScenes datasets, with attacks applied to both the Camera and LiDAR modalities.  AION achieves consistently high AUROC across all attack types and both modalities, as also shown in the [revised paper](https://openreview.net/pdf?id=SWlCJab9gZ).
>
> The results in the revised paper demonstrate strong and consistent robustness against a wide range of temporal misalignment attacks on KITTI and nuScenes, achieving high average AUROC for camera-only (0.9493) and LiDAR-only (0.9495) attacks, while sustaining robust performance under joint cross-modal attacks (0.9195 on most attacks). These results demonstrate that AION remains robust across a comprehensive suite of delay patterns and datasets.
>
> Furthermore, to validate the compatibility of AION with real-world settings, we designed a hardware-in-the-loop automotive Ethernet testbed, which serves as a practical abstraction of real in-vehicle networks. In this setup, multiple Raspberry Pis act as camera and LiDAR sensor nodes, with another Pi as the fusion node, all connected through production-grade automotive Ethernet using media converters and a switch. The entire pipeline runs over ROS2 middleware for data distribution, allowing us to model real sensor message timing and fusion workloads as deployed in production-grade systems. Within this environment, we implemented temporal misalignment attacks on the ROS2 message streams and observed AION's seamless ability to detect them.  We provided a figure of the testbed setup in Appendix F of the [revised paper](https://openreview.net/pdf?id=SWlCJab9gZ), and we will add further details on the test experiment in the final revision of the paper.

---

> ### Author Response · Authors · 2025-11-24
> **Author Response (Part 2)**
>
> **Weakness 3: Thresholding..**
>
> **Response**: Thank you for the insightful comment regarding the practical deployment of AION in real-world settings. For deployment, we adopt a single, percentile-based global threshold, calibrated using benign data only. We acknowledge that although DTW returns the optimal alignment path reward, in practice, the maximum similarity will be less than 1, depending on the scene and driving context. To remove this scene- and context-specific variance and make the threshold robust, global, and *generalized*, we scale each of the similarity matrices to [0, 1], focusing solely on the temporal alignment path, rather than scene-specific limitations. Under benign scenarios, even if discrepancies (not misalignment) exist between modalities due to benign external factors, the diagonal elements still attain their maximum value and scale approximately to 1 ($S_{ij} \approx 1$, where $i=j$). Concretely, we collect scores on benign segments and set a global threshold at a high percentile (e.g., 99th), directly targeting a desired worst-case FPR and enabling consistent deployment across sequences or fleets. In our updated experiments, we will report Precision, Recall, F1, and TPR at target FPRs.
>
>  Here are some preliminary results:
>
> *TPR at FPR<0.01*
> | Attack Type | KITTI ||| nuScenes |||
> |:------------|:-----:|:-----:|:-----:|:--------:|:-----:|:-----:|
> | | Both | Camera | Lidar | Both | Camera | Lidar |
> | Constant | 0.0600 | 0.5400 | 0.5800 | 0.0033 | 0.4908 | 0.4942 |
> | Random | 0.3385 | 0.4000 | 0.6000 | 0.3688 | 0.3704 | 0.3488 |
> | Reversal | 1.0000 | 0.4000 | 0.5200 | 0.0092 | 0.3538 | 0.3458 |
> | Jitter | 0.3036 | 0.4167 | 0.3125 | 0.2942 | 0.2486 | 0.2913 |
> | Burst | 0.2742 | 0.0333 | 0.2333 | 0.3307 | 0.3444 | 0.3333 |
> | Drift | 0.6441 | 0.5500 | 0.5500 | 0.4221 | 0.3708 | 0.3656 |
> | Scheduler | 0.2963 | 0.7143 | 0.6000 | 0.3580 | 0.3643 | 0.3732 |
>
> These metrics capture window-level detection performance, where some windows may show only subtle misalignment during a single attack event, naturally reducing per-window TPR; in the final paper, we will also include event-level results to demonstrate timely and comprehensive attack detection.
>
> **Weakness 3: Sensitivity to window size w beyond w=5 choice**
>
> **Response:** We thank for reviewers for the comments on broadening the scope of the evaluation of AION. In that line, we extended the evaluation for this rebuttal from two different perspectives: i) different temporal sampling techniques, and ii) different window sizes.
>
> - *Temporal Sampling Strategies:* Temporal sampling determines which past feature representations are selected from the historical feature queue to form the similarity matrix with a window of size \(w\). In this rebuttal, we have extended our analysis to include two additional strategies:
>   - *Factor-(k) sampling*: Selecting features at a fixed stride (e.g., every $k$-th feature)
>   - *Exponential sampling*: Placing denser samples near the current time and sparser samples further in the past
> These complementary schemes offer different temporal resolutions of the past, enabling a more comprehensive and robust evaluation. In our updated results, exponential sampling consistently delivers the strongest robustness and detection performance.
> - *Window Size Sensitivity:* For each sampling scheme, we sweep the window size $w \in \{3, 5, 7, 10\}$. We observe that moderate windows ($w = 3$ or $5$) provide the best detection performance across datasets. Larger windows (\(w \ge 7\)) yield diminishing returns. Overall, this extended evaluation shows that AION’s performance is robust to the choice of $w$ as long as the window is not excessively long. We will update the paper with the evaluation data from this analysis.
>
>
> **Question 4: Robustness to Adaptive Adversaries**
>
> **Response:** We thank the reviewer for highlighting the adaptive adversary scenario. Our current threat model assumes the attacker cannot modify sensor payloads and can only introduce timestamp manipulations. Fully adaptive attacks would require precise knowledge of AION's internal embeddings and DTW parameters, which is beyond the scope of your considered threat model for real-world AV systems. On the other hand, an adaptive attack should be regarded as a closed-loop attack. In our threat model, the attacker is only present on the in-vehicle network/sensor node, but not on the fusion node that runs AION. Closed-loop adaptive attacks (where the attacker adapts based on system feedback) would require the attacker to have real-time access to AION's detection outputs, which is unrealistic under our assumed threat model, where the attacker can only manipulate timestamps in transit but cannot access the fusion node's internal state or detection results. However, for the camera-ready version, we plan to discuss potential cases of timestamp-only adaptive attackers and discuss the potential mitigation under this extended threat scenario.

---

### Official Review · Reviewer_A4m8 · 2025-10-31

**Soundness:** 4
**Presentation:** 3
**Contribution:** 3
**Rating:** 6
**Confidence:** 2

**Summary:**

This paper looks at a very concrete hole in current multimodal AD pipelines: if an attacker only tampers with timestamps (not pixels/points) to force the ROS2/DDS synchronizer to pair camera frame t with LiDAR frame t−k, fusion degrades badly, but most existing defenses don’t look at the temporal axis. The authors propose AION, a plug-in module that (1) learns a shared camera–LiDAR representation with a continuity-aware contrastive loss so temporally close pairs stay close and far pairs stay apart, and then (2) at run time builds a cross-modal similarity matrix over a short window and runs DTW to see whether the optimal path still follows the diagonal. On KITTI and nuScenes, under synthetic constant / random delays, AION gets AUROC around 0.92–0.98 while keeping the module lightweight.

**Strengths:**

a) Fits AD architecture. The threat model (attacker sits on ROS2 / in-vehicle network and rewrites timestamps) matches how actual AD stacks glue sensors together.

b) Problem is real. Recent works have shown that temporal desync alone can tank fusion AP, but most defenses assume honest timestamps and focus on spoofing / spatial inconsistency / context violations. AION directly targets the temporal gap, which is a good angle.

c) CACL is a reasonable tweak. Using relaxed/graded negatives based on temporal distance is a sensible extension of ReCo-style contrastive learning to sequential sensor data, not just random pairs.

**Weaknesses:**

a) Attack model is fairly simple. All attacks are synthetic constant/random delays with a fixed window and fixed injection interval. Real attackers could do pattern-shaped delays, scene-aware delays, or imitate low-speed / stationary AD scenarios the paper itself says are hard to detect; for these, DTW over a short window may not separate benign vs. attack that cleanly.

b) Stationary / low-dynamics scenes remain hard. The paper admits that if both modalities are delayed by the same amount or the scene stays visually/geometrically similar, the similarity map still looks diagonal, so AION can’t tell benign from malicious. This is exactly the case an attacker in city driving would want to exploit. This limitation should be foregrounded more.

**Questions:**

a) In deployment, is the threshold global (one value for all scenes / weather / motion) or per-sequence? How sensitive was AUROC to the threshold estimated on “benign” data?
﻿
b) For nuScenes where you aggregate 6 cameras -> BEV -> shared space, do you observe different DTW patterns than in KITTI (single front camera)? In other words, is AION implicitly assuming similar FoV on both modalities?

---

> ### Author Response · Authors · 2025-11-24
> **Author Response (Part 1)**
>
> We sincerely thank Reviewer A4m8 for the constructive and encouraging review, as well as the valuable suggestions for improvement. Several of the recommended experiments and analyses have already been integrated into the revised paper, while the remainder will be included in the next update. Detailed responses to each of your points are provided below.
>
> ---
>
> **Weakness a: Attack model is fairly simple. All attacks are synthetic constant/random delays with a fixed window and fixed injection interval.**
>
> **Response:** We appreciate the concern about the simplicity of our initial attack model. To address concerns about limited attack types and robustness to real-world network behaviors, we have significantly expanded our attack suite beyond Constant and Random to include seven distinct attack types that comprehensively model various real-world failure scenarios:
>
> - *Constant*: Complete frame freezing, simulating dropped frames
> - *Random*: Random frame replacements, simulating corrupted frames
> - *Reversal*: Temporal order reversal, simulating out-of-order packet delivery
> - *Jitter*: Stochastic timing delays with probabilistic jitter, simulating network jitter
> - *Burst*: Intermittent freezes with gaps, simulating bursty network congestion
> - *Drift*: Gradual desynchronization, simulating clock skew and buffer buildup
> - *Scheduler*: Mimics CPU scheduler behavior (round-robin, priority-based)
>
> These attacks cover the different failure modes that an attacker can mimic, providing a comprehensive evaluation of robustness to real network behaviors.  We note that closed-loop attacks (where the attacker adapts based on system feedback) are beyond the scope of the current threat model. In our threat model, the attacker is only present on the in-vehicle network or sensor node, but not on the fusion node that runs AION. As closed-loop attacks would require the attacker to have real-time access to AION's detection outputs, which is unrealistic under our assumed threat model, where the attacker can only manipulate timestamps in transit but cannot access the fusion node's internal state or detection results.
>
>
> *Experimental Results*: We present results comparing AION against the baseline methods across all attack types on both the KITTI and nuScenes datasets, with attacks applied to both the Camera and LiDAR modalities.  AION achieves consistently high AUROC across all attack types and both modalities, as also shown in the [revised paper](https://openreview.net/pdf?id=SWlCJab9gZ).
>
> | Dataset   | Constant        | Random        | Reversal        | Jitter         | Burst          | Drift          | Scheduler      |
> |-----------|------------------|---------------|------------------|----------------|----------------|----------------|----------------|
> | KITTI     | 0.127–0.972      | 0.909–0.953   | 0.522–0.942      | 0.915–0.962    | 0.872–0.935    | 0.933–0.943    | 0.903–0.961    |
> | nuScenes  | 0.0983–0.9791    | 0.9394–0.9577 | 0.5095–0.9554    | 0.9060–0.9283  | 0.9365–0.9541  | 0.9433–0.9497  | 0.9369–0.9518  |
> ||
>
> The results in the revised paper demonstrate strong and consistent robustness against a wide range of temporal misalignment attacks on KITTI and nuScenes, achieving high average AUROC for camera-only (0.9493) and LiDAR-only (0.9495) attacks, while sustaining robust performance under joint cross-modal attacks (0.9195 on most attacks). These results demonstrate that AION remains robust across a comprehensive suite of delay patterns and datasets.
>
> Furthermore, to validate the compatibility of AION with real-world settings, we designed a hardware-in-the-loop automotive Ethernet testbed, which serves as a practical abstraction of real in-vehicle networks. In this setup, multiple Raspberry Pis act as camera and LiDAR sensor nodes, with another Pi as the fusion node, all connected through production-grade automotive Ethernet using media converters and a switch. The entire pipeline runs over ROS2 middleware for data distribution, allowing us to model real sensor message timing and fusion workloads as deployed in production-grade systems. Within this environment, we implemented temporal misalignment attacks on the ROS2 message streams and observed AION's seamless ability to detect them.  We provided a figure of the testbed setup in Appendix F of the [revised paper](https://openreview.net/pdf?id=SWlCJab9gZ), and we will add further details on the test experiment in the final revision of the paper.

---

> ### Author Response · Authors · 2025-11-24
> **Author Response (Part 2)**
>
> **Weakness b: Stationary / Low-Dynamics Scenes**
>
> **Response:** Thanks for the insightful comment. We acknowledge that stationary and low-dynamics scenes present a fundamental challenge for temporal misalignment detection, and we have foregrounded this limitation more prominently in our updated discussion. When both modalities are delayed by the same amount, the cross-modal similarity matrix remains diagonal under both benign and attack conditions, making detection inherently difficult. One particular mitigation would be to augment AION with Intra-/inter-modal feature flow checks, which monitor feature changes over time to detect sudden abnormal changes in the scene. While incorporating additional motion cues from CAN or IMU sensors is the ultimate promising remedy for further distinguishing truly stationary scenes from carefully crafted equal-delay attacks, such fusion with camera–LiDAR data is non-trivial and introduces several multifaceted challenges. As a result, we consider this direction beyond the scope of the current work, but view it as an important avenue for future research.
>
>
> **Question a: Threshold Selection and Sensitivity**
>
> **Response:** Thank you for the insightful comment regarding the practical deployment of AION in real-world settings. For deployment, we adopt a single, percentile-based global threshold, calibrated using benign data only. We acknowledge that although DTW returns the optimal alignment path reward, in practice, the maximum similarity will be less than 1, depending on the scene and driving context, which may result in different ranges of anomaly scores across various contexts. To remove this scene- and context-specific variance and make the threshold robust, global, and *generalized*, we scale each of the similarity matrices to [0, 1], focusing solely on the temporal alignment path, rather than scene-specific limitations. Under benign scenarios, even if discrepancies (not misalignment) exist between modalities due to benign external factors, the diagonal elements still attain their maximum value and scale approximately to 1 ($S_{ij} \approx 1$, where $i=j$). Concretely, we collect scores on benign segments and set a global threshold at a high percentile (e.g., 99th), directly targeting a desired worst-case FPR and enabling consistent deployment across sequences or fleets. In our updated experiments, we will report Precision, Recall, F1, and TPR at target FPRs.
>
> Here are some preliminary results:
>
> *TPR at FPR<0.01*
> | Attack Type | KITTI ||| nuScenes |||
> |:------------|:-----:|:-----:|:-----:|:--------:|:-----:|:-----:|
> | | Both | Camera | Lidar | Both | Camera | Lidar |
> | Constant | 0.0600 | 0.5400 | 0.5800 | 0.0033 | 0.4908 | 0.4942 |
> | Random | 0.3385 | 0.4000 | 0.6000 | 0.3688 | 0.3704 | 0.3488 |
> | Reversal | 1.0000 | 0.4000 | 0.5200 | 0.0092 | 0.3538 | 0.3458 |
> | Jitter | 0.3036 | 0.4167 | 0.3125 | 0.2942 | 0.2486 | 0.2913 |
> | Burst | 0.2742 | 0.0333 | 0.2333 | 0.3307 | 0.3444 | 0.3333 |
> | Drift | 0.6441 | 0.5500 | 0.5500 | 0.4221 | 0.3708 | 0.3656 |
> | Scheduler | 0.2963 | 0.7143 | 0.6000 | 0.3580 | 0.3643 | 0.3732 |
>
> These metrics capture window-level detection performance, where some windows may show only subtle misalignment during a single attack event, naturally reducing per-window TPR; in the final paper, we will also include event-level results to demonstrate timely and comprehensive attack detection.
>
> **Question a: FoV Mismatches and DTW Patterns (nuScenes vs KITTI)**
>
> **Response:** Thanks for this interesting comment.  From our experiment, we observe that even when the camera and LiDAR fields of view differ, the multimodal representation encoder (MRE) focuses on regions with overlapping semantics and high cross-modal similarity. In nuScenes, both camera and LiDAR provide 360° coverage, so the MRE operates on the full BEV features from both modalities when projecting into the shared representation space. In KITTI, the camera covers only the front view, while the LiDAR is 360°. During training, the MRE naturally learns to emphasize the portion of the LiDAR BEV that aligns with the camera's front-view BEV and down-weight non-overlapping regions.
>
> *Key observation*: Despite the different FoV configurations, we observe consistent DTW patterns across both datasets. In both settings, the resulting cross-modal similarity matrix remains strongly diagonal under benign conditions, with off-diagonal deviations clearly indicating temporal misalignment. This demonstrates that AION does not implicitly assume similar FoV on both modalities—the learned representation naturally adapts to the available sensor coverage, and the DTW-based detection mechanism remains effective regardless of FoV mismatches.
>
> The consistent performance across both KITTI (single front camera) and nuScenes (6-camera aggregation) further validates that AION's detection mechanism is robust to different sensor configurations and FoV arrangements.

---

### Author Response · Authors · 2025-12-02

We sincerely thank all four reviewers for their thorough and constructive feedback. The reviews have been instrumental in significantly strengthening our paper. We are encouraged that Reviewers **ApBr, A4m8**, and **hrs4** recommended acceptance, and found the contributions novel.  Below, we summarize the key points raised across all reviews and our responses.

---
**Key Strengths (Acknowledged by Reviewers):**
- **Threat Model & Practical Relevance (A4m8, hrs4, ApBr):** Multiple reviewers recognized that our threat model—timestamp manipulation to cause misaligned fusion—addresses a concrete and realistic attack surface in production AD systems.

- **Continuity-Aware Contrastive Learning (A4m8, hrs4, ApBr):** Reviewers appreciated the graded negative weighting approach as a sensible extension of contrastive learning to sequential sensor data.

- **Lightweight Plug-in Design (hrs4, ApBr):** The task-agnostic, non-intrusive architecture with low computational overhead was noted as a practical strength.

- **Strong Experimental Results (A4m8, hrs4, ApBr):** Initial AUROC of 0.92–0.98 on KITTI and nuScenes was recognized as promising.
---
**Key Weaknesses (Addressed During Rebuttal):**
- **Limited Attack Suite (A4m8, hrs4, ApBr):** Expanded from 2 to **7 distinct attack types** (Constant, Random, Reversal, Jitter, Burst, Drift, Scheduler) with comprehensive evaluation. Results show consistent robustness: average AUROC of 0.9493 (camera-only), 0.9495 (LiDAR-only), and 0.9195 (joint attacks) across all attack types.

- **Stationary/Low-Dynamics Scenes (A4m8, hrs4):** We have foregrounded this limitation more prominently in the revised paper and discussed potential mitigations (intra/inter-modal feature flow checks, CAN/IMU fusion) as future research directions.

- **Threat Model Realism (hBGi):** We clarify that in production systems, sensor payloads are cryptographically protected (DDS-Security, MACsec), while timestamps rely on insecure protocols (e.g., gPTP), making timestamp-only manipulation a realistic, low-cost attack vector.

- **Real-World Validation (hBGi, hrs4, ApBr):** We have designed and implemented a **hardware-in-the-loop testbed** using Raspberry Pis connected via production-grade automotive Ethernet and ROS2 middleware. AION successfully detects temporal misalignment attacks in this realistic environment.

- **Threshold Selection & Deployment Details (A4m8, hrs4, ApBr):** We adopt a percentile-based global threshold with similarity matrix normalization to [0,1] for scene-agnostic deployment and report **TPR at FPR<0.01** across all attack types.

- **Limited Experimental Evidence (hBGi, ApBr):** We have significantly expanded evaluation: computational overhead (3.26 ms/inference, 1.97M parameters, 42.5 MB GPU memory), window size sensitivity analysis (w ∈ {3, 5, 7, 10}), temporal sampling strategies (uniform, factor-(k), exponential).

- **Baseline Comparisons (ApBr):** We have implemented and evaluated timestamp-sanity models and sliding-window correlation detectors, both of which consistently perform weaker than AION.
---
**Summary of Major Changes:**
The following summarizes the major improvements made during the rebuttal phase. Items already incorporated into the [revised paper](https://openreview.net/pdf?id=SWlCJab9gZ) (highlighted in blue) are listed with their locations, while items marked with $^{\dagger}$ are already discussed in the rebuttal and will be added to the final version:

1. Expanded attack suite: 2 → 7 attack types with varying attack configuration (Table 1, Section 4.3, Appendix A)
2. Hardware-in-the-loop validation with ROS2 (Appendix F)
3. Attack visualization: Visualization of seven attack scenarios (Section 5.1, Figure 2, Figure 3, Appendix D, Appendix E)
4. Clarified limitations: Stationary scenes and threat model scope (Section 5.1)

5. Detection performance & scalability: Detailed results with new attacks and scalability analysis (Section 5.2, Figure 4, Table 2)

6. Enhanced evaluation$^{\dagger}$: Window size sensitivity and temporal sampling strategies

7. Deployment metrics$^{\dagger}$: TPR at practical FPRs and threshold calibration methodology

8. Baseline comparisons$^{\dagger}$: Timestamp-sanity and correlation-based detectors

---
The revised paper and the responses address all major concerns raised by the reviewers through expanded experiments, real-world validation, and clearer discussion of limitations. We believe these improvements significantly strengthen the paper's contribution to securing multimodal fusion in autonomous driving.

We note that Reviewer **ApBr** (one of three reviewers with a positive score) maintained their score of 8 during rebuttal, while Reviewer **hBGi** (the only reviewer with a negative score) remained unresponsive despite our comprehensive responses during the active discussion window. We sincerely thank the AC for their time and effort and respectfully request that they consider this context in the final assessment.

---

### Meta-Review · Area_Chair_bNDG · 2026-01-07

**Summary:**

Reviewer concerns primarily focused on the realism and diversity of the attack model, robustness in challenging scenarios (e.g., stationary or low-dynamics scenes), deployment-relevant evaluation (thresholding, latency, and computational overhead), and real-world applicability. These concerns were largely addressed by expanding the attack suite from simple constant/random delays to seven diverse and realistic temporal misalignment attack patterns, with comprehensive evaluation on both KITTI and nuScenes. The authors further strengthened the paper with clarified threat-model justification, additional baseline comparisons, detailed threshold calibration and sensitivity analyses, and explicit reporting of computational overhead and runtime costs. Real-world feasibility was partially validated through a hardware-in-the-loop testbed, though full on-vehicle deployment remains future work. Limitations in stationary and low-dynamics scenes were explicitly acknowledged and discussed, but remain fundamentally challenging under the current design.

**Reviewer Concerns:**

Reviewer A4m8:

Addressed: The reviewer’s main concerns regarding the simplicity of the attack model were addressed by substantially expanding the attack suite from constant/random delays to seven diverse and realistic attack patterns, with comprehensive evaluation on KITTI and nuScenes. The authors provide additional details on threshold selection, sensitivity, and deployment behavior.
Partially Addressed:  Limitations in stationary and low-dynamics scenes were explicitly acknowledged and discussed, but remain fundamentally challenging under the current design.

Reviewer hrs4:

Addressed: The reviewer’s concerns about the simplicity of the synthetic attack model were addressed by substantially expanding the evaluation to seven realistic temporal misalignment attack patterns and reporting consistent robustness across KITTI and nuScenes. Threshold calibration and deployment details were clarified through percentile-based global thresholding and additional sensitivity analyses on window size and temporal sampling strategies.

Partially addressed: Real-world feasibility was partially validated through a hardware-in-the-loop testbed. Limitations in stationary and low-dynamics scenes were explicitly acknowledged and discussed, but remain fundamentally challenging under the current design.

Reviewer hBGi:

Addressed: Concerns regarding the realism of the threat model were addressed through a detailed clarification explaining why timestamp-only manipulation is a realistic and practical attack vector in modern automotive systems, supported by discussion of gPTP vulnerabilities and protected sensor payloads. Experimental evidence was expanded, including a broader attack suite modeling real-world network failures, additional baseline comparisons reflecting deployed synchronization heuristics, and detailed analyses of computational overhead and scalability.

Partially addressed: Real-world validation was strengthened via a hardware-in-the-loop testbed, but full on-vehicle experiments were not conducted.

Reviewer ApBr:

Addressed:  The reviewer’s concerns regarding limited and synthetic attack scenarios were fully addressed by expanding the evaluation to seven temporal misalignment attack patterns and validating robustness across KITTI and nuScenes. Questions on deployment-relevant metrics were resolved through added analyses on threshold calibration, reporting TPR at practical FPRs, and detailed measurements of computational overhead and runtime. Baseline comparisons were strengthened by including timestamp-sanity and correlation-based detectors, and clarification was provided on the scope of model generalization and the task-agnostic nature of the plug-in design. The reviewer confirmed that all concerns were satisfactorily addressed and maintained a strong acceptance recommendation.

**Reviewer Scores:**

Reviewer A4m8 would possibly maintain their score of 6.
Reviewer hrs4 would possibly maintain their score of 6.
Reviewer hBGi would possibly raise their score from 2 to 4, with a possibility of further increase to 6.
Reviewer ApBr would possibly maintain their score of 8.

---

### Decision · Program_Chairs · 2026-01-26

Accept (Poster)